# On the Role of Noise in the Sample Complexity of Learning Recurrent Neural Networks: Exponential Gaps for Long Sequences

**Alireza F. Pour**
Cheriton School of Computer Science
University of Waterloo
`a2fathol@uwaterloo.ca`

**Hassan Ashtiani**
Department of Computing and Software
McMaster University
`zokaeiam@mcmaster.ca`

## Abstract

We consider the class of noisy multi-layered sigmoid recurrent neural networks with $w$ (unbounded) weights for classification of sequences of length $T$, where independent noise distributed according to $\mathcal{N}(0, \sigma^2)$ is added to the output of each neuron in the network. Our main result shows that the sample complexity of PAC learning this class can be bounded by $O(w \log(T/\sigma))$. For the non-noisy version of the same class (i.e., $\sigma = 0$), we prove a lower bound of $\Omega(wT)$ for the sample complexity. Our results indicate an exponential gap in the dependence of sample complexity on $T$ for noisy versus non-noisy networks. Moreover, given the mild logarithmic dependence of the upper bound on $1/\sigma$, this gap still holds even for numerically negligible values of $\sigma$.[1]

## 1 Introduction

Recurrent Neural Networks (RNNs) are effective tools for processing sequential data. They are used in numerous applications such as speech recognition (Graves et al., 2013), computer vision (Karpathy and Fei-Fei, 2015), translation (Sutskever et al., 2014), modeling dynamical systems (Hardt et al., 2018) and time series (Qin et al., 2017). Recurrent models allow us to design classes of predictors that can be applied to (i.e., take input values from) sequences of arbitrary length. For processing a sequence of $T$ elements, a predictor $f$ (e.g., a neural network) "consumes" the input elements one by one, generating an output at each step. This output is then used in the next step (as another input to $f$ along with the next element in the input sequence). Defining recurrent models formally takes some effort, and we relegate it to the next sections. In short, the function $f$ is (recursively) applied $T$ times in order to generate the ultimate outcome.

Let us fix a base class $\mathcal{F}_w$ of all multi-layered feed-forward sigmoid neural networks with $w$ weights. We can create a recurrent version of this class, which we will denote by $\text{REC}[\mathcal{F}_w, T]$, for classifying sequences of length $T$. One can study the sample complexity of PAC learning $\text{REC}[\mathcal{F}_w, T]$ with respect to different loss functions. Koiran and Sontag (1998) studied the binary-valued version of this class by applying a threshold function at the end, and proved a lower bound of $\Omega(wT)$ for its VC dimension.

There has also been efforts for proving upper bounds on the sample complexity of PAC learning $\text{REC}[\mathcal{F}, T]$ for various base classes $\mathcal{F}$ and different loss functions. Given the above lower bound, a gold standard has been achieving a linear dependence on $T$ in the upper bound. Koiran and Sontag (1998) proved an upper bound of $O(w^4 T^2)$ on the VC dimension of $\text{REC}[\mathcal{F}_w, T]$ discussed above. More recent papers have considered the more realistic setting of classification with continuous-valued

---

[1]For the full version of this paper see F. Pour and Ashtiani (2023).

37th Conference on Neural Information Processing Systems (NeurIPS 2023).

RNNs, e.g., by removing the threshold function and using a bounded Lipschitz surrogate loss. In this setting, Zhang et al. (2018) proved an upper bound of $\widetilde{O}(T^4 w\|W\|^{O(T)})$ on the sample complexity[2] where $\|W\|$ is the spectral norm of the network. Chen et al. (2020) improved over this result by proving an upper bound of $\widetilde{O}(Tw\|W\|^2 \min\{\sqrt{w}, \|W\|^{O(T)}\})$. These bounds get close to the gold standard when the spectral norm of the network satisfies $\|W\| \leq 1$.

The above upper bounds are proved by simply "unfolding" the recurrence, effectively substituting the recurrent class $\text{REC}[\mathcal{F}_w, T]$ with the (larger) class of $T$-fold compositions $\mathcal{F}_w \circ \mathcal{F}_w \ldots \circ \mathcal{F}_w$. These unfolding techniques do not exploit the fact that the function $f$ (that is applied recursively for $T$ steps to compute the output of the network) is fixed across all the $T$ steps. Consequently, the resulting sample complexity has (super-)linear dependence on $T$. Therefore, we would need a prohibitively large sample size for training recurrent models for classifying very long sequences. Nevertheless, this dependence is inevitable in light of the of lower bound of Koiran and Sontag (1998). Or is it?

In this paper, we consider a related class of *noisy* recurrent neural networks, $\text{REC}[\widetilde{\mathcal{F}_w^\sigma}, T]$. The hypotheses in this class are similar to those in $\text{REC}[\mathcal{F}_w, T]$, except that outputs of (sigmoid) activation functions are added with independent Gaussian random variables, $\mathcal{N}(0, \sigma^2)$. Our main result demonstrates that, remarkably, the noisy class can be learned with a number of samples that is only logarithmic with respect to $T$.

**Theorem 1** (Informal version of Theorem 15). *The sample complexity of PAC learning the class REC[$\widetilde{\mathcal{F}_w^\sigma}, T$] of noisy recurrent networks with respect to ramp loss is $\widetilde{O}(w \log(T/\sigma))$.*

One challenge of proving the above theorem is that the analysis involves dealing with *random* hypotheses. Therefore, unlike the usual arguments that bound the covering number of a set of deterministic maps with respect to the $\ell_2$ distance, we study the covering number of a class of random maps with respect to the total variation distance. We then invoke some of the recently developed tools in Fathollah Pour and Ashtiani (2022) for bounding these covering numbers. Another challenge is deviating from the usual "unfolding method" and exploiting the fact that in recurrent models a *fixed* function/network is applied recursively.

The mere fact that learning $\text{REC}[\widetilde{\mathcal{F}_w^\sigma}, T]$ requires less samples compared to its non-noisy counterpart is not entirely unexpected. For classification of long sequences, however, the sample complexity gap is quite drastic (i.e., exponential). We argue that a logarithmic dependency on $T$ is actually more realistic in practical situations: for finite precision machines, one can effectively break the $\Omega(T)$ barrier even for non-noisy networks. To see this, let us choose $\sigma$ to be a numerically negligible number (e.g., smaller than the numerical precision of our computing device). In this case, the class of noisy and non-noisy networks become effectively the same when implemented on a device with finite numerical precision. But then our upper bound shows a mild logarithmic dependence on $1/\sigma$.

One caveat in the above argument is that the lower bound of Koiran and Sontag (1998) is proved for the 0-1 loss and perhaps not directly comparable to the setting of the upper bound which uses a Lipcshitz surrogate loss. We address this by showing a comparable lower bound in the same setting.

**Theorem 2** (Informal version of Theorem 10). *The sample complexity of PAC learning REC[$\mathcal{F}_w, T$] with ramp loss is $\Omega(wT)$.*

In the next section we introduce our notations and define the PAC learning problem. We state the lower bound in Section 3, and the upper bound in Section 5. Sections 6, 7, and 8 provide a high-level proof of our upper bound.

**Additional Related Work.** Due to space constraints, we postpone the discussion of some additional related work to Appendix 9.

## 2 Preliminaries

### 2.1 Notations

$\|x\|_1, \|x\|_2$, and $\|x\|_\infty$ denote the $\ell_1, \ell_2$, and $\ell_\infty$ norms of a vector $x \in \mathbb{R}^d$ respectively. We denote the cardinality of a set $S$ by $|S|$. The set of natural numbers smaller or equal to $m$ is represented by

---

[2]Ignoring the dependence of the sample complexity on the accuracy and confidence parameters.

$[m]$. A vector of all zeros is denoted by $0_d = [0 \ldots 0]^\top \in \mathbb{R}^d$. We use $\mathcal{X} \subseteq \mathbb{R}^d$ as a domain set. We will study classes of vector-valued functions; a hypothesis is a Borel function $f : \mathbb{R}^d \to \mathbb{R}^p$, and a hypothesis class $\mathcal{F}$ is a set of such hypotheses.

We find it useful to have an explicit notation—here an overline—for the random versions of the above definitions: $\overline{\mathcal{X}}$ is the set of all random variables defined over $\mathcal{X}$ that admit a generalized density function[3]. $\overline{x} \in \overline{\mathcal{X}}$ is a random variable in this set. To simplify this notation, we sometimes just write $\overline{x} \in \mathbb{R}^d$ rather than $\overline{x} \in \overline{\mathbb{R}^d}$.

$\overline{y} = f(\overline{x})$ is the random variable associated with pushforward of $\overline{x}$ under Borel map $f : \mathbb{R}^d \to \mathbb{R}^p$. We use $\overline{f} : \mathbb{R}^d \to \mathbb{R}^p$ to indicate that the mapping itself is random. Random hypotheses can be applied to both random and non-random inputs—e.g., $\overline{f}(\overline{x})$ and $\overline{f}(x)$[4]. A class of random hypotheses is denoted by $\overline{\mathcal{F}}$.

**Definition 3** (Composition of Two Hypothesis Classes). *We denote by $h \circ f$ the function $h(f(x))$ (assuming the range of $f$ and the domain of $h$ are compatible). The composition of two hypothesis classes $\mathcal{F}$ and $\mathcal{H}$ is defined by $\mathcal{H} \circ \mathcal{F} = \{h \circ f \mid h \in \mathcal{H}, f \in \mathcal{F}\}$. Composition of classes of random hypotheses is defined similarly by $\overline{\mathcal{H}} \circ \overline{\mathcal{F}} = \{\overline{h} \circ \overline{f} \mid \overline{h} \in \overline{\mathcal{H}}, \overline{f} \in \overline{\mathcal{F}}\}$.*

## 2.2 Feedforward neural networks

We will first define some classes associated with feedforward neural networks. Let $\phi(x) = \frac{1}{1+e^{-x}} - \frac{1}{2}$ be the centered sigmoid function. $\Phi : \mathbb{R}^p \to [-1/2, 1/2]^p$ is the element-wise sigmoid activation function defined by $\Phi((x^{(1)}, \ldots, x^{(p)})) = (\phi(x^{(1)}), \ldots, \phi(x^{(p)}))$.

**Definition 4** (Single-Layer Sigmoid Neural Networks). *The class of single-layer sigmoid neural networks with $d$ inputs and $p$ outputs is defined by NET[d, p] = $\{f_W : \mathbb{R}^d \to [-1/2, 1/2]^p \mid f_W(x) = \Phi(W^\top x), W \in \mathbb{R}^{d \times p}\}$.*

Based on Definition 4, we can define the class of multi-layer (feedforward) neural networks (with $w$ weights) as a composition of several single-layer networks. Note that the number of hidden neurons can be arbitrary as long as the total number of weights/parameters is $w$.

**Definition 5** (Multi-Layer Sigmoid Neural Networks). *A class of multi-layer sigmoid networks with $p_0$ inputs, $p_k$ outputs, and $w$ weights that take inputs in $[-1/2, 1/2]^{p_0}$ is defined by*

$$MNET[p_0, p_k, w] = \bigcup NET[p_{k-1}, p_k] \circ \ldots \circ NET[p_0, p_1]$$

*where union is taken over all choices of $(p_1, p_2, \ldots, p_{k-1}) \in \mathbb{N}^{k-1}$ that satisfy $\sum_{i=1}^{k} p_i . p_{i-1} = w$. We say MNET[$p_0, p_k, w$] is well-defined if the union is not empty.*

Well-definedness basically means that $p_0, p_k$, and $w$ are compatible. For simplicity, in the above definition we restricted the input domain to $[-1/2, 1/2]^d$. This will help in defining the recurrent versions of these networks (since the input and output domains become compatible). However, our analysis can be easily extended to capture any bounded domain (e.g., $[-B, B]^d$).

## 2.3 Recursive application of a function and recurrent models

In this section we define REC[$\mathcal{F}, T$] which is the recurrent version of class $\mathcal{F}$ for sequences of length $T$. Let $v = (a_1, \ldots, a_m) \in \mathcal{X}^m$ for $m \in \mathbb{N}$. We define First $(v) = (a_1, \ldots, a_{m-1}) \in \mathcal{X}^{m-1}$ and Last $(v) = a_m \in \mathcal{X}$ as functions that return the first $m - 1$ and the last dimensions of the vector $v$, respectively. Let $u^{(0)}, u^{(1)}, \ldots, u^{(T-1)}$ be a sequence of inputs, where $u^{(i)} \in \mathbb{R}^p$, and let $f : \mathbb{R}^s \to \mathbb{R}^q$ be a hypothesis/mapping. In the context of recurrent models, it is useful to define the recurrent application of $f$ on this sequence. Note that out of the $q$ dimensions of the range of $f$, $q - 1$ of them are recurrent and therefore are fed back to the model. Basically, $f^R(U, t)$ will be the result of applying $f$ on the first $t$ elements of $U$ (with recurrent feedback).

---

[3]Both discrete (by using Dirac delta function) and absolutely continuous random variables admit a generalized density function.

[4]Technically, we consider $\overline{f}(x)$ to be $\overline{f}(\overline{\delta_x})$, where $\overline{\delta_x}$ is a random variable with Dirac delta measure on $x$.

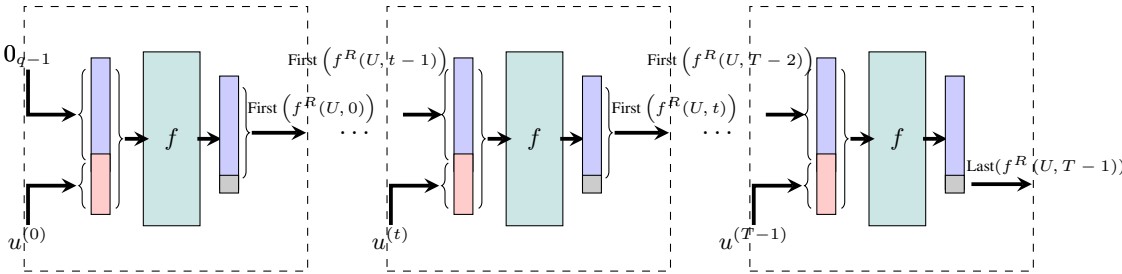

Figure 1: An example of a recurrent model in REC[$\mathcal{F}, T$]. The first $q-1$ dimensions of $f^R(U, t-1)$ is concatenated with $u^{(t)}$ to form the input at time $t$. The last dimension of $f^R(U, T-1)$ is taken to be the final output of the recurrent model.

**Definition 6** (Recurrent Application of a Function). *Let $U = \left[ u^{(0)} \ldots u^{(i)} \ldots u^{(T-1)} \right] \in \mathbb{R}^{p \times T}$ be a sequence of inputs of length $T$, where $u^{(i)} \in \mathbb{R}^p$ denotes the $i$-th column of $U$ for $0 \leq i \leq T-1$. Let $f$ be a (random) function from $\mathbb{R}^s$ to $\mathbb{R}^q$, where $s = p + q - 1$. Moreover, define $f^R(U, 0) = f\left( \begin{bmatrix} 0_{q-1} & u^{(0)} \end{bmatrix}^\top \right)$. Then, for any $1 \leq t \leq T-1$, the recursive application of $f$ is denoted by $f^R : \mathbb{R}^{p \times T} \times [T-1] \to \mathbb{R}^q$ and is defined as $f^R(U, t) = f\left( \begin{bmatrix} First\left( f^R(U, t-1) \right) & u^{(t)} \end{bmatrix}^\top \right)$.*

Now we are ready to define the (recurrent) hypothesis class REC[$\mathcal{F}, T$]. Each hypothesis in this class takes a sequence $U$ of input vectors, and applies a function $f \in \mathcal{F}$ recurrently on the elements of this sequence. The final output will be a real number. We give the formal definition in the following; also see Figure 1 for a visualization.

**Definition 7** (Recurrent Class). *Let $s, p, q \in \mathbb{N}$ such that $s = p + q - 1$. Let $\mathcal{F}$ be a class of functions from $\mathbb{R}^s$ to $\mathbb{R}^q$. The class of recurrent models with length $T$ that use functions in $\mathcal{F}$ (which we denote by recurring class) as their recurring block is defined by*

$$REC[\mathcal{F}, T] = \{ h : \mathbb{R}^{p \times T} \to \mathbb{R} \mid h(U) = Last\left( f^R(U, T-1) \right), f \in \mathcal{F} \}$$

For example, REC[MNET[$p_0, p_k, w$], $T$] is the class of (real-valued) recurrent neural networks with length $T$ that use MNET[$p_0, p_k, w$] as their recurring block. We say that REC[MNET[$p_0, p_k, w$], $T$] is *well-defined* if MNET[$p_0, p_k, w$] is well-defined and also the input/output dimensions are compatible (i.e., $p_0 \geq p_k$).

### 2.4 PAC learning with ramp loss

In this section we formulate the PAC learning model for classification with respect to the ramp loss. The use of ramp loss is natural for classification (see e.g., Boucheron et al. (2005); Bartlett et al. (2006)) and the main features of the ramp loss that we are going to exploit are boundedness and Lipschitzness. We start by introducing the ramp loss.

**Definition 8** (Ramp Loss). *Let $f : \mathcal{X} \to \mathbb{R}$ be a hypothesis and let $\mathcal{D}$ be a distribution over $\mathcal{X} \times \mathcal{Y}$. Let $(x, y) \in \mathcal{X} \times \mathcal{Y}$, where $\mathcal{Y} = \{-1, 1\}$. The ramp loss of $f$ with respect to margin parameter $\gamma > 0$ is defined as $l_\gamma(f, x, y) = r_\gamma(-f(x).y)$, where $r_\gamma$ is the ramp function defined by*

$$r_\gamma(x) = \begin{cases} 0 & x < -\gamma, \\ 1 + \frac{x}{\gamma} & -\gamma \leq x \leq 0 \\ 1 & x \geq 0. \end{cases}$$

**Definition 9** (Agnostic PAC Learning with Respect to Ramp Loss). *We say that a hypothesis class $\mathcal{F}$ of functions from $\mathcal{X}$ to $\mathbb{R}$ is agnostic PAC learnable with respect to ramp loss with margin parameter $\gamma > 0$ if there exists a learner $\mathcal{A}$ and a function $m : (0, 1)^2 \to \mathbb{N}$ with the following property: For every distribution $\mathcal{D}$ over $\mathcal{X} \times \{-1, 1\}$ and every $\epsilon, \delta \in (0, 1)$, if $S$ is a set of $m(\epsilon, \delta)$ i.i.d. samples from $\mathcal{D}$, then with probability at least $1 - \delta$ (over the randomness of $S$) we have*

$$\mathbb{E}_{(x,y) \sim \mathcal{D}} \left[ l_\gamma(\mathcal{A}(S), x, y) \right] \leq \inf_{f \in \mathcal{F}} \mathbb{E}_{(x,y) \sim \mathcal{D}} \left[ l_\gamma(\mathcal{A}(S), x, y) \right] + \epsilon.$$

The *sample complexity* of PAC learning $\mathcal{F}$ with respect to ramp loss is denoted by $m_{\mathcal{F}}(\epsilon, \delta)$, which is the minimum number of samples required for learning $\mathcal{F}$ (among all learners $\mathcal{A}$). The definition of agnostic PAC learning with respect to ramp loss works for any value of $\gamma$ and when we are analyzing the sample complexity we consider it to be a fixed constant.

## 3 A lower bound for sample complexity of learning recurrent neural networks

In this section, we consider the sample complexity of PAC learning sigmoid recurrent neural networks with respect to ramp loss. Particularly, we state a lower bound on the sample complexity of the class REC[MNET[$p_0, p_k, w$], $T$] of all sigmoid recurrent neural networks with length $T$ that use multi-layer neural networks with $w$ weights as their recurring block. The main message is that this sample complexity grows at least linearly with $T$.

**Theorem 10** (Sample Complexity Lower Bound for Recurrent Neural Networks). *For every $T \geq 3$ and $w \geq 19$ there exists a well-defined class $\mathcal{H}_w = REC[MNET[p_0, p_k, w], T]$ and a universal constant $C > 0$ such that for every $\epsilon, \delta \in (0, 1/40)$ we have*

$$m_{\mathcal{H}_w}(\epsilon, \delta) \geq C. \left( \frac{wT + \log(1/\delta)}{\epsilon^2} \right).$$

The proof of the above lower bound is based on a similar result due to Sontag et al. (1998). However, the argument in Sontag et al. (1998) is for PAC learning with respect to 0-1 loss. To extend this result for the ramp loss, we construct a binary-valued class $\mathcal{F}_w = \{f : f(U) = \text{sign}(h(U)), h \in \mathcal{H}_w\}$ where $\text{sign}(x) = 1$ if $x \geq 0$ and $\text{sign}(x) = -1$ if $x < 0$. We prove that every function $f \in \mathcal{F}_w$ can be related to another function $h \in \mathcal{H}_w$ such that the ramp loss of $h$ is almost equal to the zero-one loss of $f$. This is formalized in the following lemma, which is a key result in proving Theorem 10. The proof of Theorem 10 and Lemma 11 can be found in Appendix **??**.

**Lemma 11.** *Let $\mathcal{H}_w = REC[MNET[p_0, p_k, w], T]$ be a well-defined class and let $\mathcal{F}_w = \{f : [-1/2, 1/2]^{p \times T} \to \{-1, 1\} \mid f(U) = sign(h(U)), h \in \mathcal{H}_w\}$. Then, for every distribution $\mathcal{D}$ over $[-1/2, 1/2]^{p \times T} \times \{-1, 1\}$, $\eta > 0$, and every function $f \in \mathcal{F}_w$ there exists a function $h \in \mathcal{H}_w$ such that $\mathbb{E}_{(U,y) \sim \mathcal{D}}[l_\gamma(h, U, y)] \leq \mathbb{E}_{(U,y) \sim \mathcal{D}}[l^{0-1}(f, U, y)] + \eta$ where $l^{0-1}(f, U, y) = 1\{f(U) \neq y\}$.*

## 4 Noisy recurrent neural networks

In this section, we will define classes of noisy recurrent neural networks. Let us first define the singleton Gaussian noise class, which contains a single additive Gaussian noise function.

**Definition 12** (The Gaussian Noise Class). *The $d$-dimensional noise class with scale $\sigma \geq 0$ is denoted by $\overline{\mathcal{G}_{\sigma,d}} = \{\overline{g_{\sigma,d}}\}$. Here, $\overline{g_{\sigma,d}} : \mathbb{R}^d \to \mathbb{R}^d$ is a random function defined by $\overline{g_{\sigma,d}}(\overline{x}) = \overline{x} + \overline{z}$, where $\overline{z} \sim \mathcal{N}(\mathbf{0}, \sigma^2 I_d)$. When it is clear from the context we drop $d$ and write $\overline{\mathcal{G}_\sigma} = \{\overline{g_\sigma}\}$.*

The following is the noisy version of multi-layer networks in Definition 5. Basically, Gaussian noise is composed (Definition 3) before each layer.

**Definition 13** (Noisy Multi-Layer Sigmoid Neural Networks). *The class of all noisy multi-layer sigmoid networks with $w$ weights that take values in $[-1/2, 1/2]^{p_0}$ as input and output values in $[-1/2, 1/2]^{p_k}$ is defined by*

$$\overline{MNET_\sigma}[p_0, p_k, w] = \bigcup NET[p_{k-1}, p_k] \circ \ldots \circ \overline{\mathcal{G}_\sigma} \circ NET[p_1, p_2] \circ \overline{\mathcal{G}_\sigma} \circ NET[p_0, p_1] \circ \overline{\mathcal{G}_\sigma},$$

*where $\sigma \geq 0$ is scale of the Gaussian noise and the union is taken over all choices of $(p_1, p_2, \ldots, p_{k-1}) \in \mathbb{N}^{k-1}$ that satisfy $\sum_{i=1}^{k} p_i.p_{i-1} = w$.*

Similar to the deterministic case, $\overline{\text{MNET}_\sigma}[p_0, p_k, w]$ is said to be well-defined if the union is not empty (i.e., $p_0, p_k$ and $w$ are compatible). We can use Definition 7 to create recurrent versions of the above class. For example, REC[$\overline{\text{MNET}_\sigma}[p_0, p_k, w], T$] is a class of recurrent (and random) hypotheses for sequence of length $T$ that use $\overline{\text{MNET}_\sigma}[p_0, p_k, w]$ as their recurring block. Again, similar to the deterministic case, we say REC[$\overline{\text{MNET}_\sigma}[p_0, p_k, w], T$] is well-defined if $p_0, p_k$ and $w$ are compatible and $\overline{\text{MNET}_\sigma}[p_0, p_k, w]$ is well-defined.

## 5 PAC learning noisy recurrent neural networks

In section 3, we established an $\Omega(T)$ lower bound on the sample complexity of learning recurrent networks (i.e., REC[MNET[$p_0, p_k, w$], $T$]). In this section, we consider a related class (based on noisy recurrent neural networks) and show that the dependence of sample complexity on $T$ is only $O(\log T)$. In particular, $\overline{\mathcal{G}_\sigma} \circ \text{REC}[\overline{\text{MNET}_\sigma}[p_0, p_k, w], T]$ can be regarded as a (noisy) sibling of REC[MNET[$p_0, p_k, w$], $T$]. Since it is more standard to define PAC learnability for deterministic hypotheses, we define the deterministic version of the above class by derandomization[5].

**Definition 14** (Derandomization by Expectation). *Let $\mathcal{F}$ be a class of (random) functions from $\mathbb{R}^{p \times T}$ to $\mathbb{R}^q$. The derandomization of a function class $\overline{\mathcal{F}}$ by expectation is defined as $\mathcal{E}(\overline{\mathcal{F}}) = \{h : \mathbb{R}^{p \times T} \to \mathbb{R}^q \mid h(u) = \mathbb{E}_{\overline{f}}\left[\,\overline{f}(u)\right], \overline{f} \in \overline{\mathcal{F}}\}$.*

We show that, contrary to Theorem 10, the sample complexity of PAC learning the (derandomized) class of noisy recurrent neural networks, $\mathcal{E}(\overline{\mathcal{G}_\sigma} \circ \text{REC}[\overline{\text{MNET}_\sigma}[p_0, p_k, w], T])$, grows at most logarithmically with $T$ while it still enjoys the same linear dependence on $w$. This is formalized in the following theorem (see Appendix **??** for a proof).

**Theorem 15** (Main Result). *Let $\overline{\mathcal{Q}_w} = \overline{\mathcal{G}_\sigma} \circ REC[\overline{MNET_\sigma}[p_0, p_k, w], T]$ be any well-defined class and assume $T \in \mathbb{N}, 0 < \sigma < 1, \epsilon, \delta \in (0, 1)$. Then the sample complexity of learning $\mathcal{H}_w = \mathcal{E}(\overline{\mathcal{Q}_w})$ is upper bounded by*

$$m_{\mathcal{H}_w}(\epsilon, \delta) = O\left(\frac{w \log\left(\frac{wT}{\epsilon\sigma} \log\left(\frac{wT}{\epsilon\sigma}\right)\right) + \log(1/\delta)}{\epsilon^2}\right) = \widetilde{O}\left(\frac{w \log\left(\frac{T}{\sigma}\right) + \log(1/\delta)}{\epsilon^2}\right),$$

*where $\widetilde{O}$ hides logarithmic factors.*

One feature of the above theorem is the mild logarithmic dependence on $1/\sigma$. Therefore, we can take $\sigma$ to be numerically negligible and still get a significantly smaller sample complexity compared to the deterministic case for large $T$. Note that adding such small values of noise would not change the empirical outcome of RNNs on finite precision computers.

The milder (logarithmic) dependency on $T$ is achieved by a novel analysis that involves bounding the covering number of noisy recurrent networks with respect to the total variation distance. Also, instead of "unfolding" the network, we exploit the fact that the same function/hypothesis is being used recurrently. We also want to emphasize that the above bound does not depend on the norms of weights of the network. Achieving this is challenging, since a little bit of noise in a previous layer can change the output of the next layer drastically. The next few sections are dedicated to give a high-level proof of this theorem.

## 6 Covering numbers: the classical view

One of the main tools to derive sample complexity bounds for learning a class of functions is studying their covering numbers. In this section we formalize this classic tool.

**Definition 16** (Covering Number). *Let $(\mathcal{X}, \rho)$ be a metric space. A set $A \subset \mathcal{X}$ is $\epsilon$-covered by a set $C \subseteq A$ with respect to $\rho$, if for all $a \in A$ there exists $c \in C$ such that $\rho(a, c) \leq \epsilon$. We denote by $N(\epsilon, A, \rho)$ the cardinality of the smallest set $C$ that $\epsilon$-covers $A$ and we refer to is as the $\epsilon$-covering number of $A$ with respect to metric $\rho$.*

The notion of covering number is defined with respect to a metric $\rho$. We now give the definition of extended metrics, which we will use to define *uniform* covering numbers. The extended metrics can be seen as measures of distance between two hypotheses on a given input set.

**Definition 17** (Extended Metrics). *Let $(\mathcal{X}, \rho)$ be a metric space. Let $u = (a_1, \ldots, a_m), v = (b_1, \ldots, b_m) \in \mathcal{X}^m$ for $m \in \mathbb{N}$. The $\infty$-extended and $\ell_2$-extended metrics over $\mathcal{X}^m$ are defined by $\rho^{\infty, m}(u, v) = \sup_{1 \leq i \leq m} \rho(a_i, b_i)$ and $\rho^{\ell_2, m}(u, v) = \sqrt{\frac{1}{m} \sum_{i=1}^m (\rho(a_i, b_i))^2}$, respectively. We drop $m$ and use $\rho^\infty$ or $\rho^{\ell_2}$ if it is clear from the context.*

---

[5]One can also define PAC learnability for a class of random hypotheses and get a similar result without taking the expectation. However, working with a deterministic class helps to contrast the result with that of Theorem 10.

A useful property about extended metrics is that the $\infty$-extended metric always upper bounds the $\ell_2$-extended metric, i.e., $\rho^{\ell_2}(u,v) \leq \rho^{\infty}(u,v)$ for all $u,v \in \mathcal{X}$. Based on the above definition of extended metrics, we define the uniform covering number of a hypothesis class with respect to $\|.\|_2$.

**Definition 18** (Uniform Covering Number with Respect to $\|.\|_2$). *Let $\mathcal{F}$ be a hypothesis class of functions from $\mathcal{X}$ to $\mathcal{Y}$. For a set of inputs $S = \{x_1, x_2, \ldots, x_m\} \subseteq \mathcal{X}$, we define the restriction of $\mathcal{F}$ to $S$ as $\mathcal{F}_{|S} = \{(f(x_1), f(x_2), \ldots, f(x_m)) : f \in \mathcal{F}\} \subseteq \mathcal{Y}^m$. The uniform $\epsilon$-covering numbers of hypothesis class $\mathcal{F}$ with respect to $\|.\|_2^{\infty}, \|.\|_2^{\ell_2}$ are denoted by $N_U(\epsilon, \mathcal{F}, m, \|.\|_2^{\infty})$ and $N_U(\epsilon, \mathcal{F}, m, \|.\|_2^{\ell_2})$ and are the maximum values of $N(\epsilon, \mathcal{F}_{|S}, \|.\|_2^{\infty,m})$ and $N(\epsilon, \mathcal{F}_{|S}, \|.\|_2^{\ell_2,m})$ over all $S \subseteq \mathcal{X}$ with $|S| = m$, respectively.*

The following theorem connects the notion of uniform covering number with PAC learning. It converts a bound on the $\|.\|_2^{\ell_2}$ uniform covering number of a hypothesis class to a bound on the sample complexity of PAC learning the class; see Appendix **??** for a more detailed discussion.

**Theorem 19.** *Let $\mathcal{F}$ be a class of functions from $\mathcal{X}$ to $\mathbb{R}$. Then there exists an algorithm $\mathcal{A}$ with the following property: For every distribution $\mathcal{D}$ over $\mathcal{X} \times \{-1, 1\}$ and every $\epsilon, \delta \in (0,1)$, if $S$ is a set of $m$ i.i.d. samples from $\mathcal{D}$, then with probability at least $1 - \delta$ (over the randomness of $S$),*

$$\mathbb{E}_{(x,y)\sim\mathcal{D}}\left[l_\gamma\left(\mathcal{A}(S), x, y\right)\right]$$

$$\leq \inf_{f\in\mathcal{F}}\mathbb{E}_{(x,y)\sim\mathcal{D}}\left[l_\gamma\left(f, x, y\right)\right] + 16\epsilon + \frac{24}{\sqrt{m}}\sqrt{\ln N_U(\gamma\epsilon, \mathcal{F}, m, \|.\|_2^{\ell_2})} + 6\sqrt{\frac{\ln(2/\delta)}{2m}}.$$

*Moreover, the algorithm that returns the function with the minimum error on $S$ satisfies the above property (i.e., Algorithm $\mathcal{A}$ such that $\mathcal{A}(S) = \arg\min_{f\in\mathcal{F}}\frac{1}{|S|}\sum_{(x,y)\in S}l_\gamma\left(f, x, y\right)$).*

# 7 Total variation covers for random hypotheses

One idea to prove a generalization bound for noisy neural networks is to bound their covering numbers. However, noisy neural networks are random functions, and therefore their behaviours on a sample set cannot be directly compared. Instead, one can compare the output distributions of a random function on two sample sets. We therefore use the recently developed tools from Fathollah Pour and Ashtiani (2022) to define and study covering numbers for random hypotheses. These covering numbers are defined based on metrics between distributions. Specifically, our analysis is based on the notion of uniform covering number with respect to total variation distance.

**Definition 20** (Total Variation Distance). *Let $\mu$ and $\nu$ denote two probability measures over $\mathcal{X}$ and let $\Omega$ be the Borel sigma-algebra over $\mathcal{X}$. The TV distance between $\mu$ and $\nu$ is defined by*

$$d_{TV}(\mu, \nu) = \sup_{B\in\Omega}|\mu(B) - \nu(B)|.$$

*Furthermore, if $\mu$ and $\nu$ have densities $f$ and $g$ then*

$$d_{TV}(\mu, \nu) = \sup_{B\in\Omega}\left|\int_B(f(x) - g(x))dx\right| = \frac{1}{2}\int_{\mathcal{X}}|f(x) - g(x)|\,dx = \frac{1}{2}\|f - g\|_1.$$

For two random variables $\overline{x}$ and $\overline{y}$ with probability measures $\mu$ and $\nu$ we sometimes abuse the notation and write $d_{TV}(\overline{x}, \overline{y})$ instead of $d_{TV}(\mu, \nu)$. For example, we write $d_{TV}(\overline{f_1}(\overline{x}), \overline{f_2}(\overline{x}))$ in order to refer to the Total Variation (TV) distance between pushforwards of $\overline{x}$ under mappings $\overline{f_1}$ and $\overline{f_2}$. We also write $d_{TV}^{\infty,m}\left(\left(\overline{f_1}(\overline{x_1}), \ldots, \overline{f_1}(\overline{x_m})\right), \left(\overline{f_2}(\overline{x_1}), \ldots, \overline{f_2}(\overline{x_m})\right)\right)$ to refer to the extended TV distance between mappings of the set $S = \{\overline{x_1}, \ldots, \overline{x_m}\}$ by $\overline{f_1}$ and $\overline{f_2}$. We use the extended total variation distance to define the uniform covering number for classes of random hypotheses.

**Definition 21** (Uniform Covering Number for Classes of Random Hypotheses). *Let $\overline{\mathcal{F}}$ be a class of random hypotheses from $\overline{\mathcal{X}}$ to $\overline{\mathcal{Y}}$. For a set of random variables $\overline{S} = \{\overline{x_1}, \overline{x_2}, \ldots, \overline{x_m}\} \subseteq \overline{\mathcal{X}}$, the restriction of $\overline{\mathcal{F}}$ to $\overline{S}$ is defined as $\overline{\mathcal{F}}_{|\overline{S}} = \{(\overline{f}(\overline{x_1}), \overline{f}(\overline{x_2}), \ldots, \overline{f}(\overline{x_m})) : \overline{f} \in \overline{\mathcal{F}}\} \subseteq \overline{\mathcal{Y}}^m$. Let $\Gamma \subseteq \overline{\mathcal{X}}$. The uniform $\epsilon$-covering numbers of $\overline{\mathcal{F}}$ with respect to $\Gamma$ and $d_{TV}^{\infty}$ is defined by*

$$N_U(\epsilon, \overline{\mathcal{F}}, m, d_{TV}^{\infty}, \Gamma) = \sup_{S\subseteq\Gamma, |S|=m} N(\epsilon, \overline{\mathcal{F}}_{|\overline{S}}, d_{TV}^{\infty,m}).$$

Some hypothesis classes that we analyze (e.g., single-layer noisy neural networks) may have "global" total variation covers that do not depend on $m$. This will be addressed with the following notation: $N_U(\epsilon, \overline{\mathcal{F}}, \infty, \rho^\infty, \Gamma) = \lim_{m \to \infty} N_U(\epsilon, \overline{\mathcal{F}}, m, \rho^\infty, \Gamma)$. The set $\Gamma$ in Definition 21 is used to define the input domain for which we want to find the covering number of a class of random hypotheses. For instance, some of the covers that we see are derived with respect to inputs with bounded domain or some need the input to be first smoothed by Gaussian noise. In this paper, we will be working with the following choices of $\Gamma$

- $\Gamma = \overline{\mathcal{X}_d}$ and $\Gamma = \overline{\mathcal{X}_{B,d}}$: the set of all random variables defined over $\mathbb{R}^d$ and $[-B, B]^d$, respectively, that admit a generalized density function. For example, we use $\overline{\mathcal{X}_{0.5,d}}$ to address the set of random variables in $[-1/2, 1/2]^d$.

- $\Gamma = \overline{\Delta_{p \times T}} = \{\overline{U} \mid \overline{U} = \begin{bmatrix} \overline{\delta_{u^{(0)}}} & \dots & \overline{\delta_{u^{(T-1)}}} \end{bmatrix}^\top, u^{(i)} \in \mathbb{R}^p\}$ and $\Gamma = \overline{\Delta_{B,p \times T}} = \{\overline{U} \mid \overline{U} = \begin{bmatrix} \overline{\delta_{u^{(0)}}} & \dots & \overline{\delta_{u^{(T-1)}}} \end{bmatrix}^\top, u^{(i)} \in [-B, B]^p\}$, where $\overline{\delta_{u^{(i)}}}$ is the random variable associated with Dirac delta measure on $u^{(i)}$. Note that $\overline{\Delta_{B,p \times T}} \subset \overline{\Delta_{p \times T}}$.

- $\Gamma = \overline{\mathcal{G}_{\sigma,d}} \circ \overline{\mathcal{X}_{B,d}} = \{\overline{g_{\sigma,d}}(\overline{x}) \mid \overline{x} \in \overline{\mathcal{X}_{B,d}}\}$: all members of $\overline{\mathcal{X}_{B,d}}$ after being "smoothed" by adding (convolving the density with) Gaussian noise.

We mentioned in Section 6 that a bound on the $\|.\|_2^{\ell_2}$ uniform covering number can be connected to a bound on sample complexity of PAC learning. We now show that a bound on $d_{TV}^\infty$ covering number of a class of random hypotheses can be turned into a bound on the $\|.\|_2^{\ell_2}$ covering number of its derandomized version and, thus, PAC learning it.

**Theorem 22** ($\|.\|_2^{\ell_2}$ Cover of $\mathcal{E}(\mathcal{F})$ From $d_{TV}^\infty$ Cover of $\mathcal{F}$ (Fathollah Pour and Ashtiani, 2022)). *Let $\overline{\mathcal{F}}$ be a class of functions from $\mathbb{R}^{p \times T}$ to $[-B, B]^q$. Then for every $\epsilon > 0$ and $m \in \mathbb{N}$ we have*

$$N_U(2B\epsilon\sqrt{q}, \mathcal{E}(\overline{\mathcal{F}}), m, \|.\|_2^{\ell_2}) \leq N_U(\epsilon, \overline{\mathcal{F}}, m, d_{TV}^\infty, \overline{\Delta_{p \times T}}) \leq N_U(\epsilon, \overline{\mathcal{F}}, \infty, d_{TV}^\infty, \overline{\Delta_{p \times T}}).$$

# 8 Bounding the covering number of recurrent models

In Section 6, we mentioned that finding a bound on covering number of a hypothesis class is a standard approach to bound its sample complexity. In the previous section, we introduced a new notion of covering number with respect to total variation distance that was developed by Fathollah Pour and Ashtiani (2022). We showed how this notion can be related to PAC learning for classes of random hypotheses. In the following, we give an overview of the techniques used to find a bound on the $d_{TV}^\infty$ covering number of the class of noisy recurrent models. We also discuss why this bound results in a sample complexity that has a milder logarithmic dependency on $T$, compared to bounds proved by "unfolding" the recurrence and replacing the recurrent model with the $T$-fold composition.

One advantage of analyzing the uniform covering number with respect to TV distance is that it comes with a useful composition tool. The following theorem basically states that when two classes of hypotheses have bounded TV covers, their composition class has a bounded cover too. Note that such a result does not hold for the usual definition of covering number (e.g., Definition 18); see Fathollah Pour and Ashtiani (2022) for details.

**Theorem 23** (TV Cover for Composition of Random Classes, Lemma 18 of Fathollah Pour and Ashtiani (2022)). *Let $\overline{\mathcal{F}}$ be a class of random hypotheses from $\mathbb{R}^d$ to $\mathbb{R}^p$ and $\overline{\mathcal{H}}$ be a class of random hypotheses from $\mathbb{R}^p$ to $\mathbb{R}^q$. For any $\epsilon_1, \epsilon_2 > 0$ and $m \in \mathbb{N}$, denote $N_1 = N_U\left(\epsilon_1, \overline{\mathcal{F}}, m, d_{TV}^\infty, \overline{\mathcal{X}_d}\right)$. Then we have,*

$$N_U\left(\epsilon_1 + \epsilon_2, \overline{\mathcal{H}} \circ \overline{\mathcal{F}}, m, d_{TV}^\infty, \overline{\mathcal{X}_d}\right) \leq N_U\left(\epsilon_2, \overline{\mathcal{H}}, mN_1, d_{TV}^\infty, \overline{\mathcal{X}_p}\right). N_1.$$

An approach to bound the TV uniform covering number of a recurrent model $\text{REC}[\overline{\mathcal{F}}, T]$ is to consider it as the $T$-fold composition $\overline{\mathcal{F}} \circ \overline{\mathcal{F}} \dots \circ \overline{\mathcal{F}}$. One can then use a similar analysis to that of Fathollah Pour and Ashtiani (2022) to bound the covering number of the $T$-fold composition. Unfortunately, this approach fails to capture the fact that a *fixed* function $\overline{f} \in \overline{\mathcal{F}}$ is applied recursively, and therefore results in a sample complexity bound that grows at least linearly with $T$.

Instead, we take another approach to bound the covering number of recurrent models. Intuitively, we notice that any function in the $T$-fold composite class $\overline{\mathcal{F}} \circ \dots \circ \overline{\mathcal{F}} = \{\overline{f_1} \circ \dots \circ \overline{f_T} \mid \overline{f_1}, \dots, \overline{f_T} \in$

$\overline{\mathcal{F}}$} is determined by $T$ functions from $\overline{\mathcal{F}}$. On the other hand, any function in $\text{REC}[\overline{\mathcal{F}}, T] = \left\{ \overline{h} \mid \overline{h}(U) = \text{Last}\left( \overline{f}^R(U, T-1) \right) \right\}$ is only defined by one function in $\overline{\mathcal{F}}$ and the capacity of this class must not be as large as the capacity of $\overline{\mathcal{F}} \circ \ldots \circ \overline{\mathcal{F}}$. Interestingly, data processing inequality for total variation distance (Lemma **??**) suggests that if two functions $\overline{f}$ and $\hat{\overline{f}}$ are "globally" close to each other with respect to TV distance (i.e., $d_{TV}(\overline{f}(\overline{x}), \hat{\overline{f}}(\overline{x})) \leq \epsilon$ for every $\overline{x}$ in the domain), then $d_{TV}(\overline{f}(\overline{f}(\overline{x})), \hat{\overline{f}}(\hat{\overline{f}}(\overline{x}))) \leq 2\epsilon$ (i.e., $\overline{f} \circ \overline{f}$ and $\hat{\overline{f}} \circ \hat{\overline{f}}$ are also close to each other). By applying the data processing inequality recursively, we can see that for the $T$-fold composition we have $d_{TV}(\overline{f} \circ \ldots \circ \overline{f}(\overline{x}), \hat{\overline{f}} \circ \ldots \circ \hat{\overline{f}}(\overline{x})) \leq \epsilon T$. The above approach results in the following theorem which bounds the $\epsilon$-covering number of a noisy recurrent model with respect to TV distance by the $(\epsilon/T)$-covering number of its recurring class. Intuitively, this theorem helps us to bound the covering number of noisy recurrent models using the bounds obtained for their non-recurrent versions. Here, Gaussian noise is added to both the input of the model (i.e., $\overline{\mathcal{F}}_\sigma = \overline{\mathcal{F}} \circ \overline{\mathcal{G}}_\sigma$) and the output of the model (by composing with $\overline{\mathcal{G}}_\sigma$).

**Theorem 24** (TV Covering Number of $\overline{\mathcal{G}}_\sigma \circ \text{REC}[\overline{\mathcal{F}}_\sigma, T]$ From $\overline{\mathcal{G}}_\sigma \circ \overline{\mathcal{F}}_\sigma$). *Let $s, p, q \in \mathbb{N}$ such that $s = p + q - 1$. Let $\overline{\mathcal{F}}$ be a class of functions from $\overline{\mathcal{X}_{B,s}}$ to $\overline{\mathcal{X}_{B,q}}$ and denote by $\overline{\mathcal{F}}_\sigma = \overline{\mathcal{F}} \circ \overline{\mathcal{G}}_{\sigma,s}$ the class of its composition with noise. Then we have*

$$N_U\left(\epsilon, \overline{\mathcal{G}}_\sigma \circ REC[\overline{\mathcal{F}}_\sigma, T], \infty, d_{TV}^\infty, \overline{\Delta_{B, p \times T}}\right) \leq N_U\left(\epsilon/T, \overline{\mathcal{G}}_{\sigma,q} \circ \overline{\mathcal{F}}_\sigma, \infty, d_{TV}^\infty, \overline{\mathcal{X}_{B,s}}\right).$$

For using this theorem, one needs to have a finer $\epsilon/T$-cover for the recurring class. As we will see in the next section, this will translate into a mild logarithmic sample complexity dependence on $T$.

## 8.1 Covering noisy recurrent networks

An example of $\overline{\mathcal{F}}_\sigma$ is the class $\overline{\text{MNET}}_\sigma[p_0, p_k, w]$ of well-defined noisy multi-layer networks (Definition 13). Theorem 24 suggests that a bound on the covering number of $\overline{\mathcal{G}}_\sigma \circ \text{REC}[\overline{\text{MNET}}_\sigma[p_0, p_k, w], T]$ can be found from a bound for $\overline{\mathcal{G}}_\sigma \circ \overline{\text{MNET}}_\sigma[p_0, p_k, w]$. We use the following theorem as a bound for the class of single-layer noisy sigmoid networks together with theorem 23 to bound the covering number of $\overline{\mathcal{G}}_\sigma \circ \overline{\text{MNET}}_\sigma[p_0, p_k, w]$ (see Appendix **??**, Theorem **??**).

**Theorem 25** (A TV Cover for Single-Layer Noisy Neural Networks, Theorem 25 of Fathollah Pour and Ashtiani (2022)). *For every $p, d \in \mathbb{N}, \epsilon > 0, \sigma < 5d/\epsilon$ we have*

$$\log N_U(\epsilon, \overline{\mathcal{G}}_{\sigma,p} \circ NET[d, p], \infty, d_{TV}^\infty, \overline{\mathcal{G}}_{\sigma,d} \circ \overline{\mathcal{X}_{0.5,d}}) \leq p(d+1) \log \left( 30 \frac{d^{5/2} \sqrt{\ln\left(\frac{5d - \epsilon\sigma}{\epsilon\sigma}\right)}}{\epsilon^{3/2}\sigma^2} \ln\left(\frac{5d}{\epsilon\sigma}\right) \right).$$

Interestingly, the above bound (on the logarithm of the covering number) is logarithmic with respect to $1/\epsilon$. We will extend this result to multi-layer noisy networks, and then apply Theorem 24 to obtain the following bound on the covering number noisy recurrent neural networks. Crucially, the dependency (of the logarithm of the covering number) on $T$ is only logarithmic.

**Theorem 26** (A TV Covering Number Bound for Noisy Sigmoid Recurrent Networks). *Let $T \in \mathbb{N}$. For every $\epsilon, \sigma \in (0, 1)$ and every well-defined class $REC[\overline{MNET}_\sigma[p_0, p_k, w], T]$ we have*

$$\log N_U\left(\epsilon, \overline{\mathcal{G}}_\sigma \circ REC[\overline{MNET}_\sigma[p_0, p_k, w], T], \infty, d_{TV}^\infty, \overline{\Delta_{0.5, p \times T}}\right)$$
$$= O\left( w \log\left(\frac{wT}{\epsilon\sigma} \log\left(\frac{wT}{\epsilon\sigma}\right)\right)\right) = \tilde{O}\left(w \log\left(\frac{T}{\epsilon\sigma}\right)\right).$$

Finally, we turn the above bound into a $\|.\|_2^{\ell_2}$ covering number bound for the derandomized function $\mathcal{E}\left(\overline{\mathcal{G}}_\sigma \circ \text{REC}[\overline{\text{MNET}}_\sigma[p_0, p_k, w], T]\right)$ by an application of Theorem 22. We then upper bound the sample complexity by the logarithm of covering number (see Theorem 19) and conclude Theorem 15.
**Limitations and future work.** Our results are derived for sigmoid (basically bounded, monotone, and Lipschitz) activation functions. It is open whether such results can be proved for unbounded activation functions such as RELU. Our results are theoretical and we leave empirical evaluations on the performance of noisy networks to future work.

# 9 More on related work

There is plethora of work on generalization in neural networks. There are a family of approaches that aim to bound the VC-dimension of neural networks. (Baum and Haussler, 1988; Maass, 1994; Goldberg and Jerrum, 1995; Vidyasagar, 1997; Sontag et al., 1998; Koiran and Sontag, 1998; Bartlett et al., 1998; Bartlett and Maass, 2003; Bartlett et al., 2019). These approaches result in generalization bounds that are dependent on the number of parameters. Another family of approaches are aimed at obtaining generalization bounds that are dependent on the norms of the weights and Lipschitz continuity properties of the network (Bartlett, 1996; Anthony et al., 1999; Zhang, 2002; Neyshabur et al., 2015; Bartlett et al., 2017; Neyshabur et al., 2018; Golowich et al., 2018; Arora et al., 2018; Nagarajan and Kolter, 2018; Long and Sedghi, 2020). It has been observed that these generalization bounds are usually vacuous in practice. One speculation is that the implicit bias of gradient descent (Gunasekar et al., 2017; Arora et al., 2019; Ji et al., 2020; Chizat and Bach, 2020; Ji and Telgarsky, 2021) can lead to benign overfitting (Belkin et al., 2018, 2019; Bartlett et al., 2020, 2021). It has also been conjectured that uniform convergence theory may not be able to fully capture the performance of neural networks in practice (Nagarajan and Kolter, 2019; Zhang et al., 2021). It has been shown that there are data-dependent approaches that can achieve non-vacuouys bounds (Dziugaite and Roy, 2017; Zhou et al., 2019; Negrea et al., 2019). There are also other approaches that are independent of data (Arora et al., 2018); see Fathollah Pour and Ashtiani (2022) for more details.

Adding different types of noise such as dropout noise (Srivastava et al., 2014), DropConnect (Wan et al., 2013), and Denoising AutoEncoders (Vincent et al., 2008) are shown to be helpful in training neural networks. Wang et al. (2019) and Gao and Zhou (2016) theoretically analyze the generalization under dropout noise. More recently, Fathollah Pour and Ashtiani (2022) developed a framework to study the generalization of classes of noisy hypotheses and show that adding noise to the output of neurons in a network can be helpful in generalization. Jim et al. (1996) show that additive and multiplicative noise can help speed up the convergence of RNNs on local minima surfaces. Recently, Lim et al. (2021) showed that noisy RNNs are more stable and robust to input perturbations by formalizing the regularization effects of noise.

Another line of work focuses on the generalization of neural network that are trained with Stochastic Gradient Descent (SGD) or its noisy variant Stochastic Gradient Langevin Descent (SGLD) (Russo and Zou, 2016; Xu and Raginsky, 2017; Russo and Zou, 2019; Steinke and Zakynthinou, 2020; Raginsky et al., 2017; Haghifam et al., 2020; Neu et al., 2021). Zhao et al. (2020) analyze the memory properties of recurrent networks and how well they can remember the input sequence. Tu et al. (2020) study the generalization of RNN by analyzing the Fisher-Rao norm of weights, which they obtain from the gradients of the network. They offer generalization bounds that can potentially become polynomial in $T$. Allen-Zhu and Li (2019) analyze the change in output through the dynamics of training RNNs and prove generalization bounds for recurrent networks that are again polynomial in $T$.

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
