## B Miscellaneous facts

550 **Lemma 27** (Data Processing Inequality for TV Distance). *Given two random variables $\overline{x_1}, \overline{x_2} \in \overline{\mathcal{X}}$,*
551 *and a (random) Borel function $f : \mathcal{X} \to \mathcal{Y}$,*

$$d_{TV}(f(\overline{x_1}), f(\overline{x_2})) \leq d_{TV}(\overline{x_1}, \overline{x_2}).$$

552 **Lemma 28.** *Let $\overline{x}, \overline{y} \in \overline{\mathcal{X}}$ be two random variables with probability measures $\mu$ and $\nu$. Denote*
553 *by $\Pi(\mu, \nu)$ the set of all their couplings. Then, there exists $\pi^* \in \Pi(\mu, \nu)$ such that $\mathbb{P}_{\pi^*}[\overline{x} \neq \overline{y}] =$*
554 *$d_{TV}(\mu, \nu)$, where the subscript $\pi^*$ signals that the probability law is associated with the coupling $\pi^*$.*
555 *Moreover, for any coupling $\pi \in \Pi(\mu, \nu)$ we have $\mathbb{P}_{\pi}[\overline{x} \neq \overline{y}] \geq d_{TV}(\mu, \nu)$.*

556 We use the following two lemmas to reason about the covering number of our recurrent model when
557 we take the first dimensions of the output at each time $t$ and when we concatenate new inputs with
558 the outputs. The first lemma states that if two random variables are close to each other with respect
559 to total variation distance, then they are still close after the applications of the First $(.)$ and Last $(.)$
560 functions.

561 **Lemma 29** (From TV of Random Variable to TV of First and Last). *Let $\overline{x_1}, \overline{x_2} \in \mathbb{R}^d$ be two random*
562 *variables. We have*
$$d_{TV}\left(First\left(\overline{x_1}\right), First\left(\overline{x_2}\right)\right) \leq d_{TV}\left(\overline{x_1}, \overline{x_2}\right),$$
$$d_{TV}\left(Last\left(\overline{x_1}\right), Last\left(\overline{x_2}\right)\right) \leq d_{TV}\left(\overline{x_1}, \overline{x_2}\right).$$

563 *Proof.* We know that First $(.)$ and Last $(.)$ are functions from $\mathbb{R}^d$ to $\mathbb{R}^{d-1}$. Therefore we can apply
564 Lemma 27 and conclude the result. $\square$

565 The next lemma is used to bound the total variation distance between two random variables after
566 being concatenated with the input at time $t$. In that case, we let $\overline{x_1}$ and $\overline{x_2}$ in the lemma to be
567 First $\left(f^R\left(U, t-1\right)\right)$ and First $\left(\hat{f}^R\left(U, t-1\right)\right)$, which are in $\overline{\mathcal{X}_{p_k-1}}$. We also let $\overline{y}$ be $u^{(t)} \in \overline{\Delta_d}$,
568 which is the input at time $t$.

569 **Lemma 30** (From TV of Random Variable to TV of Concatenation). *Let $\overline{x_1}, \overline{x_2}$ be random variables*
570 *in $\overline{\mathcal{X}_d}$. Further, let $\overline{y}$ a random variable in $\overline{\Delta_d}$. If we have $d_{TV}\left(\overline{x_1}, \overline{x_2}\right) \leq \epsilon$, then*

$$d_{TV}\left(\begin{bmatrix}\overline{x_1} & \overline{y}\end{bmatrix}^\top, \begin{bmatrix}\overline{x_2} & \overline{y}\end{bmatrix}^\top\right) \leq \epsilon.$$

571 *Proof.* Let $y \in \overline{\Delta_d}$ be the random variable with Dirac delta measure on $y_0$. From Lemma 28 we
572 know that there exists a maximal coupling $\pi^*$ of $\overline{x_1}$ and $\overline{x_2}$ such that $d_{TV}(\overline{x_1}, \overline{x_2}) = \mathbb{P}_{\pi^*}[\overline{x_1} \neq \overline{x_2}]$
573 and denote the density associated with $\mathbb{P}_{\pi^*}$ by $f^*$. Let $\gamma$ be a coupling of $\begin{bmatrix}\overline{x_1} & \overline{y_1}\end{bmatrix}^\top$ and $\begin{bmatrix}\overline{x_2} & \overline{y_2}\end{bmatrix}^\top$
574 such that

$$\hat{f}\left(\begin{bmatrix}x_1 & y_1\end{bmatrix}^\top, \begin{bmatrix}x_2 & y_2\end{bmatrix}^\top\right) = \begin{cases} f^*(x_1, x_2) & y_1 = y_2 = y_0, \\ 0 & \text{otherwise.} \end{cases}$$

575 We can easily verify that $\gamma$ is a valid coupling. Denote by $f_{x_1 y}$ the density of the random variable
576 $\begin{bmatrix}\overline{x_1} & y\end{bmatrix}^\top$. We know that

$$f_{\overline{x_1 y}}\left(\begin{bmatrix}x_1 & y_1\end{bmatrix}^\top\right) = \begin{cases} f_{\overline{x_1}}(x_1) & y = y_0, \\ 0 & \text{otherwise,} \end{cases}$$

577 where $f_{\overline{x_1}}$ is the density function of the random variable $\overline{x_1}$. We can observe that density associated
578 with the marginal of $\gamma$ would be the same as the density of the marginal of $\pi^*$ at points where $y = y_0$
579 and it is zero otherwise. On the other hand, we know that $\pi^*$ is a valid coupling of $\overline{x_1}$ and $\overline{x_2}$ and

therefore the density of its marginal is $f_{\overline{x_1}}$. This concludes that the density of the marginal of $\gamma$ is indeed $f_{\overline{x_1 y}}$. We can show the similar thing for the other marginal, which concludes that $\gamma$ is a valid coupling.

Therefore, from Lemma 28 we can write that

$$
\begin{aligned}
d_{TV}\left(\begin{bmatrix}\overline{x_1} & \overline{y}\end{bmatrix}^\top, \begin{bmatrix}\overline{x_2} & \overline{y}\end{bmatrix}^\top\right) &\leq \mathbb{P}_\gamma\left[\begin{bmatrix}\overline{x_1} & \overline{y}\end{bmatrix}^\top \neq \begin{bmatrix}\overline{x_2} & \overline{y}\end{bmatrix}^\top\right] \\
&\leq \int_{\begin{bmatrix}x_1\\y\end{bmatrix}\neq\begin{bmatrix}x_2\\y\end{bmatrix}} \hat{f}\left(\begin{bmatrix}x_1 & y\end{bmatrix}^\top, \begin{bmatrix}x_2 & y\end{bmatrix}^\top\right) \leq \int_{\substack{x_1\neq x_2,\\y=y_0}} \hat{f}\left(\begin{bmatrix}x_1 & y\end{bmatrix}^\top, \begin{bmatrix}x_2 & y\end{bmatrix}^\top\right) \\
&\leq \int_{\substack{x_1\neq x_2,\\y=y_0}} f^*\left(x_1, x_2\right) \leq \mathbb{P}_{\pi^*}\left[\overline{x_1} \neq \overline{x_2}\right] = d_{TV}\left(\overline{x_1}, \overline{x_2}\right) \leq \epsilon.
\end{aligned}
$$

$\square$

# C  Proof of lower bound

In order to prove Theorem 10, we first need to give the definition of PAC Learning with respect to $0-1$ loss.

**Definition 31** (Agnostic PAC Learning with Respect to 0-1 Loss)**.** *We say that a hypothesis class $\mathcal{F}$ of functions from $\mathcal{X}$ to $\mathbb{R}$ is agnostic PAC learnable with respect to $0-1$ loss if there exists a learner $\mathcal{A}$ and a function $m^{0-1} : (0,1)^2 \to \mathbb{N}$ with the following property: For every distribution $\mathcal{D}$ over $\mathcal{X} \times \{-1, 1\}$ and every $\epsilon, \delta \in (0, 1)$, if $S$ is a set of $m(\epsilon, \delta)$ i.i.d. samples from $\mathcal{D}$, then with probability at least $1 - \delta$ (over the randomness of S) we have*

$$
\mathbb{E}_{(x,y)\sim D}\left[l^{0-1}(\mathcal{A}(S), x, y)\right] \leq \inf_{f\in\mathcal{F}} \mathbb{E}_{(x,y)\sim D}\left[l^{0-1}(f, x, y)\right] + \epsilon.
$$

Same as Definition 9, we denote by $m_{\mathcal{F}}^{0-1}(\epsilon, \delta)$ the *sample complexity* of PAC learning $\mathcal{F}$ with respect to $0-1$ loss, which is the minimum number of samples required for learning $\mathcal{F}$ among all learners $\mathcal{A}$.

Before proving Theorem 10, we first prove Lemma 11, which, as mentioned before, is a core part of the proof. We state the lemma once more for completeness.

**Lemma 32.** *Let $\mathcal{H}_w = REC[MNET[p_0, p_k, w], T]$ be a well-defined class and let $\mathcal{F}_w = \{f : [-1/2, 1/2]^{p\times T} \to \{-1, 1\} \mid f(U) = sign(h(U)), h \in \mathcal{H}_w\}$. Then, for every distribution $\mathcal{D}$ over $[-1/2, 1/2]^{p\times T} \times \{-1, 1\}$, $\eta > 0$, and every function $f \in \mathcal{F}_w$ there exists a function $h \in \mathcal{H}_w$ such that $\mathbb{E}_{(U,y)\sim\mathcal{D}}\left[l_\gamma(h, U, y)\right] \leq \mathbb{E}_{(U,y)\sim\mathcal{D}}\left[l^{0-1}(f, U, y)\right] + \eta$ where $l^{0-1}(f, U, y) = 1\{f(U) \neq y\}$.*

*Proof.* We know that $\mathcal{H}_w = \{h : \mathbb{R}^{p\times T} \to [-1/2, 1/2] \mid h(u) = \text{Last}\left(b^R(U, T-1)\right), b \in \text{MNET}[p_0, p_k, w]\}$. Similarly, $\mathcal{F}_w = \{f : \mathbb{R}^{p\times T} \to \{-1, 1\} \mid f(u) = \text{sign}\left(\text{Last}\left(b^R(U, T-1)\right)\right), b \in \text{MNET}[p_0, p_k, w]\}$. Fix a distribution $\mathcal{D}$ over $[-1/2, 1/2]^{p\times T} \times \{-1, 1\}$. Define

$$
z = \min_b \arg\max_{0<x<\frac{1}{2}} \mathbb{P}\left[\left|\text{Last}\left(b^R(U, T-1)\right)\right| \geq x\right] \geq 1 - \eta,
$$

where the minimum is taken over all well-defined multi-layer neural networks $b$ in $\text{MNET}[p_0, p_k, w]$. The last dimension of function $b$ is in $[-1/2, 1/2]$ and, intuitively, $z$ is the largest possible value such that $\mathbb{P}\left[-z < \text{Last}\left(b^R(U, T-1)\right) < z\right] < \eta$.

Let $f$ be any function in $\mathcal{F}_w$ and let $b = b_{k-1} \circ \ldots \circ b_0$ be the $k$-layer network associated with $f$ where $b_i$'s are single-layer sigmoid neural networks, i.e., $f(U) = \text{sign}\left(\text{Last}\left(b^R(U, T-1)\right)\right)$. Let $W_{k-1} = \begin{bmatrix}v_1 & \ldots & v_{p_k}\end{bmatrix}^\top$ be the weight matrix associated with $b_{k-1}$. Denote by $\hat{W}_{k-1} = \begin{bmatrix}v_1 & \ldots & c.v_{p_k}\end{bmatrix}^\top$ the matrix that is exactly the same as $W_{k-1}$ but every element in its last row is multiplied by $c = \phi^{-1}(\gamma)/\phi^{-1}(z)$. Note that $z > 0$ and, therefore, $\phi^{-1}(z) > 0$. Let $\hat{b}_{k-1}$ be the single-layer neural network that is defined by weight matrix $\hat{W}_{k-1}$, i.e., $\hat{b}_{k-1}(x) = \Phi\left(\hat{W}_{k-1}^\top x\right)$. Denote $\hat{b} = \hat{b}_{k-1} \circ \ldots \circ b_0$ and let $h(U) = \text{Last}\left(\hat{b}^R(U, T-1)\right)$ for any $U \in \mathbb{R}^{p\times T}$. Clearly, $\hat{b} \in \text{MNET}[p_0, p_k, w]$

and $h \in \mathcal{H}_w$. We claim that $\mathbb{E}_{(U,y)\sim\mathcal{D}}\left[l_\gamma\left(h, U, y\right)\right] \leq \mathbb{E}_{(U,y)\sim\mathcal{D}}\left[l^{0-1}\left(f, U, y\right)\right] + \eta$. We can write the definition of ramp loss as

$$
\begin{aligned}
\mathbb{E}_{(U,y)\sim\mathcal{D}}\left[l_\gamma\left(h, U, y\right)\right] &= \mathbb{E}_{(U,y)\sim\mathcal{D}}\left[r_\gamma\left(-h(U).y\right)\right] \\
&= \mathbb{E}_{(U,y)\sim\mathcal{D}}\left[r_\gamma\left(-h(U).y\right)\middle|\,|h(U)| \geq \phi\left(c.\phi^{-1}\left(z\right)\right)\right].\mathbb{P}\left[|h(U)| \geq \phi\left(c.\phi^{-1}\left(z\right)\right)\right] \\
&+ \mathbb{E}_{(U,y)\sim\mathcal{D}}\left[r_\gamma\left(-h(U).y\right)\middle|\,|h(U)| < \phi\left(c.\phi^{-1}\left(z\right)\right)\right].\mathbb{P}\left[|h(U)| < \phi\left(c.\phi^{-1}\left(z\right)\right)\right] \quad (1) \\
&= \mathbb{E}_{(U,y)\sim\mathcal{D}}\left[r_\gamma\left(-h(U).y\right)\middle|\,|h(U)| \geq \gamma\right].\mathbb{P}\left[|h(U)| \geq \gamma\right] \\
&+ \mathbb{E}_{(U,y)\sim\mathcal{D}}\left[r_\gamma\left(-h(U).y\right)\middle|\,|h(U)| < \gamma\right].\mathbb{P}\left[|h(U)| < \gamma\right],
\end{aligned}
$$

where we used the fact that sigmoid is a monotonic increasing function with a unique inverse and that $\phi\left(c.\phi^{-1}(z)\right) = \phi\left(\phi^{-1}(\gamma)\right) = \gamma$. Notice that whenever $|h(U)| \geq \gamma$ we can also conclude that either $h(U).y \geq \gamma$ or $h(U).y \leq -\gamma$. This means that $r_\gamma\left(-h(U).y\right)$ is either 0 or 1. When $h(U).y \geq \gamma$ we have $r_\gamma\left(-h(U).y\right) = 0$ and when $h(U).y \leq -\gamma$ we have $r_\gamma\left(-h(U).y\right) = 1$. In other words if $|h(U)| \geq \gamma$, we have

$$r_\gamma\left(-h(U).y\right) = 1\left\{\text{sign}\left(h(U)\right) \neq y\right\} \quad (2)$$

On the other hand, we know that $\gamma, z > 0$ and $c = \phi^{-1}(\gamma)/\phi^{-1}(z) > 0$. Consequently, $\text{sign}\left(h(U)\right) = \text{sign}\left(\text{Last}\left(\hat{b}^R\left(U, T-1\right)\right)\right) = f(U)$ for any $U \in \mathbb{R}^{p\times T}$. Lemma 36 suggests that

$$\mathbb{P}\left[\left|\text{Last}\left(b^R(U, T-1)\right)\right| < z\right] = \mathbb{P}\left[\left|\text{Last}\left(\hat{b}^R\left(U, T-1\right)\right)\right| < \phi\left(c.\phi^{-1}(z)\right)\right] = \mathbb{P}\left[|h(U)| < \gamma\right].$$

Moreover, we know that $z$ is chosen such that $\mathbb{P}\left[\left|\text{Last}\left(b^R(U, T-1)\right)\right| < z\right] < \eta$ and the ramp loss is at most 1. Taking this and Equations 1 and 2 into account we can write that

$$
\begin{aligned}
\mathbb{E}_{(U,y)\sim\mathcal{D}}\left[l_\gamma\left(h, U, y\right)\right] &= \mathbb{E}_{(U,y)\sim\mathcal{D}}\left[1\left\{\text{sign}\left(h(U)\right) \neq y\right\}\middle|\,|h(U)| \geq \gamma\right].\mathbb{P}\left[|h(U)| \geq \gamma\right] \\
&+ \mathbb{E}_{(U,y)\sim\mathcal{D}}\left[r_\gamma\left(-h(U).y\right)\middle|\,|h(U)| < \gamma\right].\mathbb{P}\left[\left|\text{Last}\left(b^R(U, T-1)\right)\right| < z\right] \\
&\leq \mathbb{E}_{(U,y)\sim\mathcal{D}}\left[1\left\{\text{sign}\left(h(U)\right) \neq y\right\}\middle|\,|h(U)| \geq \gamma\right].\mathbb{P}\left[|h(U)| \geq \gamma\right] + \eta \\
&\leq \mathbb{E}_{(U,y)\sim\mathcal{D}}\left[1\left\{\text{sign}\left(h(U)\right) \neq y\right\}\middle|\,|h(U)| \geq \gamma\right].\mathbb{P}\left[|h(U)| \geq \gamma\right] \\
&+ \mathbb{E}_{(U,y)\sim\mathcal{D}}\left[1\left\{\text{sign}\left(h(U)\right) \neq y\right\}\middle|\,|h(U)| < \gamma\right].\mathbb{P}\left[|h(U)| < \gamma\right] + \eta \\
&\leq \mathbb{E}_{(U,y)\sim\mathcal{D}}\left[1\left\{\text{sign}\left(h(U)\right) \neq y\right\}\right] + \eta \\
&\leq \mathbb{E}_{(U,y)\sim\mathcal{D}}\left[l^{0-1}\left(f, U, y\right)\right] + \eta.
\end{aligned}
$$

$\square$

## Proof of Theorem 10.

*Proof.* Define $\mathcal{F}_w = \left\{f : [-1/2, 1/2] \to \{-1, 1\} \mid f(U) = \text{sign}\left(h(U)\right), h \in \mathcal{H}_w\right\}$ as the class of all sigmoid recurrent networks with $w$ weights that output binary values. Let $\mathcal{D}$ be a distribution over $[-1/2, 1/2]^{p\times T} \times \{-1, 1\}$. From Lemma 11 we know that for every $f \in \mathcal{F}_w$ there exists a function $h \in \mathcal{H}_w$ such that $\mathbb{E}_{(U,y)\sim\mathcal{D}}\left[l_\gamma\left(h, U, y\right)\right] \leq \mathbb{E}_{(U,y)\sim\mathcal{D}}\left[l^{0-1}\left(f, U, y\right)\right] + \eta$, where $\eta > 0$ is any small value. Therefore, we can write that

$$\inf_{h\in\mathcal{H}_w}\mathbb{E}_{(U,y)\sim\mathcal{D}}\left[l_\gamma\left(h, U, y\right)\right] \leq \inf_{f\in\mathcal{F}_w}\mathbb{E}_{(U,y)\sim\mathcal{D}}\left[l^{0-1}\left(f, U, y\right)\right] + \eta. \quad (3)$$

Let $m_{\mathcal{H}_w}(\epsilon, \delta)$ denote the sample complexity of PAC learning $\mathcal{H}_w$ with respect to ramp loss. Therefore, there exists an algorithm $\mathcal{A}$ that receives a set $S$ of $m \geq m_{\mathcal{H}_w}(\epsilon, \delta)$ i.i.d. samples from $\mathcal{D}$ and returns $\hat{h} = \mathcal{A}(S)$ such that with probability at least $1 - \delta$ we have

$$\mathbb{E}_{(U,y)\sim\mathcal{D}}\left[l^\gamma\left(\hat{h}, U, y\right)\right] \leq \inf_{h\in\mathcal{H}_w}\mathbb{E}_{(U,y)\sim\mathcal{D}}\left[l^\gamma\left(h, U, y\right)\right] + \epsilon.$$

Let $\hat{f}$ be a function in $\mathcal{F}_w$ such that $\hat{f}(U) = \text{sign}\left(\hat{h}(U)\right)$ for every $U \in [-1/2, 1/2]^{p\times T}$. Given the definitions of $0-1$ loss and ramp loss, it is easy to verify that $\mathbb{E}_{(U,y)\sim\mathcal{D}}\left[l\right]^{0-1}\left(\hat{f}, U, y\right) \leq$

639    $\mathbb{E}_{(U,y)\sim\mathcal{D}}\left[l\right]^{\gamma}\left(\hat{h},U,y\right)$. Taking this and Equation 3 into account, we can define a new algorithm $\mathcal{A}'$

640    that, given the set $S$, returns $\hat{f}\in\mathcal{F}_w$ such that with probability at least $1-\delta$ we have

$$\mathbb{E}_{(U,y)\sim\mathcal{D}}\left[l\right]^{0-1}\left(\hat{f},U,y\right)\le\inf_{h\in\mathcal{H}_w}\mathbb{E}_{(U,y)\sim\mathcal{D}}\left[l\right]^{\gamma}\left(h,U,y\right)+\epsilon\le\inf_{f\in\mathcal{F}_w}\mathbb{E}_{(U,y)\sim\mathcal{D}}\left[l\right]^{0-1}\left(f,U,y\right)+\epsilon+\eta.$$

641    This means that we have

$$m_{\mathcal{F}_w}^{0-1}(\epsilon+\eta,\delta)\le m_{\mathcal{H}_w}(\epsilon,\delta). \tag{4}$$

642    On the other hand, from Theorem 34 we now that the VC-dimension of $\mathcal{F}_w$ is $\Omega(wT)$. Moreover,

643    Theorem 33 suggests that

$$m_{\mathcal{F}_w}^{0-1}(\epsilon,\delta)=\Omega\left(\frac{wT+\log(1/\delta)}{\epsilon^2}\right).$$

644    Taking the above equation and Equation 4 into account, by setting $\eta=O(\epsilon)$, we can write that

$$m_{\mathcal{H}_w}(\epsilon,\delta)=\Omega\left(\frac{wT+\log(1/\delta)}{\epsilon^2}\right),$$

645    which concludes our result.      $\square$

646    The following theorem states that we can find a lower bound on the sample complexity of PAC

647    learning $\mathcal{F}$ with respect to $0-1$ loss based on its VC-dimension. For a proof see Theorems 5.2, and

648    5.10 in Anthony et al. (1999).

649    **Theorem 33** (Lower Bound on the Sample Complexity of PAC Learning (Anthony et al., 1999))**.**

650    *Let $\mathcal{F}$ be a class of functions from a domain $\mathcal{X}$ to $\{-1,1\}$ and let $d=VC(\mathcal{F})$ be the VC-dimension*

651    *of the class $\mathcal{F}$. Assume $d<\infty$. Then there exists an absolute constant $C$ such that for every*

652    $(\epsilon,\delta)\in(0,1/40)$ *we have*

$$m_{\mathcal{F}}^{0-1}(\epsilon,\delta)\ge C\frac{d+\log(1/\delta)}{\epsilon^2}.$$

653    We now introduce a lower bound on the VC-dimension of sigmoid recurrent neural networks with

654    binary outputs which is based on a result due to Koiran and Sontag (1998).

655    **Theorem 34** (A Lower Bound on VC-Dimension of Sigmoid Recurrent Neural Networks)**.** *For*

656    *every $T\ge 3$ and $w\ge 19$ there exists a well-defined class $\mathcal{H}_w=REC[MNET[p_0,p_k,w],T]$ with*

657    *the following property: The VC-dimension of $\mathcal{F}_w=\{f:[-1/2,1/2]^{p\times T}\to\{-1,1\}\mid f(U)=$*

658    *$sign\left(h(U)\right),h\in\mathcal{H}_w\}$ is $\Omega\left(wT\right)$.*

659    The proof of the above theorem is essentially the same as the proof of the result in Koiran and Sontag

660    (1998). The only difference is that the we should construct our network in a way that the last two

661    dimensions of the output of MNET$[p_0,p_k,w]$ must be similar to each other in order to feed back

662    the value of Last $\left(b^R(f,t-1)\right)$ with an extra node. Therefore, we only need a network that has a

663    constant factor more weights than the network that is proposed in Koiran and Sontag (1998) which

664    does not change the order of sample complexity.

665    **C.1    Lemmas used in the proof of Lemma 11**

666    In the following, we state two lemmas that will help in proving Lemma 11.

667    **Lemma 35.** *Let $W_{k-1}=[v_1\ldots v_{p_k}]\in\mathbb{R}^{p_{k-1}\times p_k}$ and $\hat{W}_{k-1}=\begin{bmatrix}v_1 & \ldots & c.v_{p_k}\end{bmatrix}^{\top}$ for a constant*

668    *$c>0$. Define two single-layer networks $b_{k-1}(x)=\Phi\left(W_{k-1}^{\top}x\right)$ and $\hat{b}_{k-1}(x)=\Phi\left(\hat{W}_{k-1}^{\top}x\right)$. Then,*

669    *for any two multi-layer networks $b=b_{k-1}\circ\ldots\circ b_0$ and $\hat{b}=\hat{b}_{k-1}\circ\ldots\circ b_0$ in a well-defined class*

670    *MNET$[p_0,p_k,w]$, every $U\in[-1/2,1/2]^{p\times T}$, and every $t\in[T-1]$ we have*

$$First\left(b^R\left(U,t\right)\right)=First\left(\hat{b}^R\left(U,t\right)\right).$$

*Proof.* We prove by induction. Denote $r = b_{k-2} \circ \ldots \circ b_o$. Therefore, we have $b = b_{k-1} \circ r$ and $\hat{b} = \hat{b}_{k-1} \circ r$. For $t = 0$, we can denote $x^{(0)} = r\left(\begin{bmatrix} 0_{q-1} & u^{(0)} \end{bmatrix}^\top\right)$ and write that

$$
\text{First}\left(b^R\left(U, 0\right)\right) = \text{First}\left(b_{k-1}\left(r\left(\begin{bmatrix} 0_{q-1} \\ u^{(0)} \end{bmatrix}\right)\right)\right) = \text{First}\left(b_{k-1}\left(x^{(0)}\right)\right)
$$

$$
= \text{First}\left(\begin{bmatrix} \phi\left(\langle v_1, x^{(0)} \rangle\right) \\ \vdots \\ \phi\left(\langle v_{p_k-1}, x^{(0)} \rangle\right) \end{bmatrix}\right) = \text{First}\left(\begin{bmatrix} \phi\left(\langle v_1, x^{(0)} \rangle\right) \\ \vdots \\ \phi\left(\langle c.v_{p_k-1}, x^{(0)} \rangle\right) \end{bmatrix}\right)
$$

$$
= \text{First}\left(\hat{b}_{k-1}\left(x^{(0)}\right)\right) = \text{First}\left(\hat{b}^R\left(U, 0\right)\right),
$$

where $\langle v_i, x^{(t)} \rangle$ denotes the inner product between vectors $v_i$ and $x^{(t)}$. Assume that we have $\text{First}\left(b^R\left(U, t-1\right)\right) = \text{First}\left(\hat{b}^R\left(U, t-1\right)\right)$ for $t-1 \in [T-2]$. We now prove that we also have $\text{First}\left(b^R\left(U, t\right)\right) = \text{First}\left(\hat{b}^R\left(U, t\right)\right)$. Denote $x^{(t)} = r\left(\begin{bmatrix} \text{First}\left(b^R\left(U, t-1\right)\right) & u^{(t)} \end{bmatrix}^\top\right)$. We can then write that

$$
\text{First}\left(b^R\left(U, t\right)\right) = \text{First}\left(b_{k-1} \circ r\left(\begin{bmatrix} \text{First}\left(b^R\left(U, t-1\right)\right) \\ u^{(t)} \end{bmatrix}\right)\right) = \text{First}\left(b_{k-1}\left(x^{(t)}\right)\right)
$$

$$
= \text{First}\left(\begin{bmatrix} \phi\left(\langle v_1, x^{(t)} \rangle\right) \\ \vdots \\ \phi\left(\langle v_{p_k-1}, x^{(t)} \rangle\right) \end{bmatrix}\right) = \text{First}\left(\begin{bmatrix} \phi\left(\langle v_1, x^{(t)} \rangle\right) \\ \vdots \\ \phi\left(\langle c.v_{p_k-1}, x^{(t)} \rangle\right) \end{bmatrix}\right)
$$

$$
= \text{First}\left(\hat{b}_{k-1}\left(x^{(t)}\right)\right) = \text{First}\left(\hat{b}^R\left(U, t\right)\right).
$$

$\square$

**Lemma 36.** *Let $W_{k-1} = [v_1 \ldots v_{p_k}] \in \mathbb{R}^{p_{k-1} \times p_k}$ and $\hat{W}_{k-1} = \begin{bmatrix} v_1 & \ldots & c.v_{p_k} \end{bmatrix}^\top$ for a constant $c > 0$. Define two single-layer networks $b_{k-1}(x) = \Phi\left(W_{k-1}^\top x\right)$ and $\hat{b}_{k-1}(x) = \Phi\left(\hat{W}_{k-1}^\top x\right)$. Let $\mathcal{D}$ be a distribution over $[-1/2, 1/2]^{p \times T}$. Then, for any two multi-layer networks $b = b_{k-1} \circ \ldots \circ b_0$ and $\hat{b} = \hat{b}_{k-1} \circ \ldots \circ b_0$ in a well-defined class MNET[$p_0, p_k, w$] we have*

$$
\mathbb{P}\left[\left|Last\left(b^R\left(U, T-1\right)\right)\right| < z\right] = \mathbb{P}\left[\left|Last\left(\hat{b}^R\left(U, T-1\right)\right)\right| < \phi\left(c.\phi^{-1}\left(z\right)\right)\right],
$$

*where $\phi^{-1}(z)$ is the inverse of sigmoid function $\phi$ at $z$.*

*Proof.* Denote $r = b_{k-2} \circ \ldots \circ b_o$ and $x^{(T-1)} = r\left(\begin{bmatrix} \text{First}\left(b^R\left(U, T-2\right)\right) & u^{(T-1)} \end{bmatrix}^\top\right)$. Note that

$$
\text{Last}\left(b^R\left(U, T-1\right)\right) = \text{Last}\left(b_{k-1} \circ r\left(\begin{bmatrix} \text{First}\left(b^R\left(U, T-2\right)\right) \\ u^{(T-1)} \end{bmatrix}\right)\right)
$$

$$
= \text{Last}\left(b_{k-1}\left(x^{(T-1)}\right)\right) = \phi\left(\langle v_{p_k}, x^{(T-1)} \rangle\right),
$$

where $\langle v_{p_k}, x^{(T-1)} \rangle$ denotes the inner product between $v_{p_k}$ and $x^{(T-1)}$. From Lemma 35, we know that $\text{First}\left(b^R\left(U, T-2\right)\right) = \text{First}\left(\hat{b}^R\left(U, T-2\right)\right)$. Therefore, we also have that

$$
\text{Last}\left(\hat{b}^R\left(U, T-1\right)\right) = \text{Last}\left(\hat{b}_{k-1} \circ r\left(\begin{bmatrix} \text{First}\left(\hat{b}^R\left(U, T-2\right)\right) \\ u^{(T-1)} \end{bmatrix}\right)\right)
$$

$$
= \text{Last}\left(\hat{b}_{k-1}\left(x^{(T-1)}\right)\right) = \phi\left(\langle c.v_{p_k}, x^{(T-1)} \rangle\right).
$$

Considering the above equations and the facts that $\phi(x)$ is an invertible and strictly increasing function and that $\phi(x) = -\phi(-x)$, we can write

$$\mathbb{P}\left[\left|\text{Last}\left(b^R\left(U, T-1\right)\right)\right| < z\right] = \mathbb{P}\left[-z < \text{Last}\left(b^R\left(U, T-1\right)\right) < z\right]$$

$$= \mathbb{P}\left[-z \le \phi\left(\langle v_{p_k}, x^{(T-1)}\rangle\right) < z\right] = \mathbb{P}\left[\phi^{-1}(-z) < \langle v_{p_k}, x^{(T-1)}\rangle < \phi^{-1}(z)\right]$$

$$= \mathbb{P}\left[-c.\phi^{-1}(z) \le \langle c.v_{p_k}, x^{(T-1)}\rangle < c.\phi^{-1}(z)\right]$$

$$= \mathbb{P}\left[\phi\left(-c.\phi^{-1}(z)\right) < \phi\left(\langle c.v_{p_k}, x^{(T-1)}\rangle\right) < \phi\left(c.\phi^{-1}(z)\right)\right]$$

$$= \mathbb{P}\left[-\phi\left(c.\phi^{-1}(z)\right) < \phi\left(\langle c.v_{p_k}, x^{(T-1)}\rangle\right) < \phi\left(c.\phi^{-1}(z)\right)\right]$$

$$= \mathbb{P}\left[\left|\text{Last}\left(\hat{b}^R\left(U, T-1\right)\right)\right| < \phi\left(c.\phi^{-1}(z)\right)\right].$$

$\square$

# D   Proof of upper bound

## D.1   Proof of Theorem 24

We prove the following general theorem which holds for input domains $\overline{\mathcal{X}_s}$ and $\Delta_{p \times T}$.

**Theorem 37** (TV Covering Number of $\overline{\mathcal{G}_\sigma} \circ \text{REC}[\overline{\mathcal{F}_\sigma}, T]$ From $\overline{\mathcal{G}_\sigma} \circ \overline{\mathcal{F}_\sigma}$). *Let $s, p, q \in \mathbb{N}$ such that $s = p + q - 1$. Let $\overline{\mathcal{F}}$ be a class of functions from $\overline{\mathcal{X}_s}$ to $\overline{\mathcal{X}_q}$ and denote by $\overline{\mathcal{F}_\sigma} = \overline{\mathcal{F}} \circ \mathcal{G}_{\sigma,s}$ the class of its composition with noise. Then we have*

$$N_U\left(\epsilon, \overline{\mathcal{G}_\sigma} \circ \text{REC}[\overline{\mathcal{F}_\sigma}, T], \infty, d_{TV}^\infty, \overline{\Delta_{p \times T}}\right) \le N_U\left(\epsilon/T, \overline{\mathcal{G}_{\sigma,q}} \circ \overline{\mathcal{F}_\sigma}, \infty, d_{TV}^\infty, \overline{\mathcal{X}_s}\right).$$

*Proof.* Let $C = \{\overline{g_{\sigma,q}} \circ \hat{f}_i \circ \overline{g_{\sigma,s}} \mid \hat{f}_i \circ \overline{g_{\sigma,s}} \in \overline{\mathcal{F}_\sigma}, i \in [r]\}$ be a global $\epsilon$-cover for $\overline{\mathcal{G}_{\sigma,s}} \circ \overline{\mathcal{F}_\sigma}$ with respect to domain $\overline{\mathcal{X}_s}$ and $d_{TV}^\infty$. Therefore, $|C| \le N_U\left(\epsilon, \overline{\mathcal{G}_{\sigma,q}} \circ \overline{\mathcal{F}_\sigma}, \infty, d_{TV}^\infty, \overline{\mathcal{X}_s}\right)$. Then for any function $\overline{g_{\sigma,q}} \circ f \circ \overline{g_{\sigma,s}} \in \overline{\mathcal{G}_{\sigma,q}} \circ \overline{\mathcal{F}_\sigma}$ we know that there exists a function $\overline{g_{\sigma,q}} \circ \hat{f}_i \circ \overline{g_{\sigma,s}}$ in $C$ such that for every $\overline{x} \in \overline{\mathcal{X}_s}$ we have $d_{TV}\left(\overline{g_{\sigma,q}} \circ f \circ \overline{g_{\sigma,s}}(\overline{x}), \overline{g_{\sigma,q}} \circ \hat{f}_i \circ \overline{g_{\sigma,s}}(\overline{x})\right) \le \epsilon$. Denote $\overline{h} = f \circ \overline{g_{\sigma,s}}$ and $\overline{\hat{h}_i} = \hat{f}_i \circ \overline{g_{\sigma,s}}$. We prove by induction that for any input matrix $\overline{U} = \begin{bmatrix} \overline{u^{(0)}} & \cdots & \overline{u^{(T-1)}} \end{bmatrix} \in \overline{\Delta_{p \times T}}$, where $\overline{u^{(t)}} = \overline{\delta_{u^{(t)}}}$, we have $d_{TV}\left(\overline{g_{\sigma,q}} \circ \overline{h}^R\left(\overline{U}, T-1\right), \overline{g_{\sigma,q}} \circ \overline{\hat{h}_i}^R\left(\overline{U}, T-1\right)\right) \le T\epsilon$.

We start by proving that $d_{TV}\left(\overline{g_{\sigma,q}} \circ \overline{h}^R\left(\overline{U}, 0\right), \overline{g_{\sigma,q}} \circ \overline{\hat{h}_i}^R\left(\overline{U}, 0\right)\right) \le \epsilon$. Denote $\overline{x^{(0)}} = \begin{bmatrix} \overline{\delta_{0_{q-1}}} & \overline{u^{(0)}} \end{bmatrix}^\top \in \overline{\Delta_s}$. We can write that

$$d_{TV}\left(\overline{g_{\sigma,q}} \circ h^R\left(\overline{U}, 0\right), \overline{g_{\sigma,q}} \circ \hat{h}_i^R\left(\overline{U}, 0\right)\right)$$

$$= d_{TV}\left(\overline{g_{\sigma,q}} \circ f \circ \overline{g_{\sigma,s}}\left(\begin{bmatrix} \overline{\delta_{0_{q-1}}} \\ u^{(0)} \end{bmatrix}\right), \overline{g_{\sigma,q}} \circ \hat{f}_i \circ \overline{g_{\sigma,s}}\left(\begin{bmatrix} \overline{\delta_{0_{q-1}}} \\ u^{(0)} \end{bmatrix}\right)\right).$$

Since $\left(\begin{bmatrix} \overline{\delta_{0_{q-1}}} & \overline{u^{(0)}} \end{bmatrix}^\top\right) \in \overline{\mathcal{X}_s}$ and considering the fact that $\overline{g_{\sigma,q}} \circ f \circ \overline{g_{\sigma,s}} = \overline{g_{\sigma,q}} \circ h \in \overline{\mathcal{G}_{\sigma,q}} \circ \overline{\mathcal{F}_\sigma}$ and $\overline{g_{\sigma,q}} \circ \hat{f}_i \circ \overline{g_{\sigma,s}} = \overline{g_{\sigma,q}} \circ \overline{\hat{h}_i} \in \overline{\mathcal{G}_{\sigma,q}} \circ \overline{\mathcal{F}_\sigma}$ are globally $\epsilon$-close over $\overline{\mathcal{X}_s}$, we get that

$$d_{TV}\left(\overline{g_{\sigma,q}} \circ \overline{h}^R\left(\overline{U}, 0\right), \overline{g_{\sigma,q}} \circ \overline{\hat{h}_i}^R\left(\overline{U}, 0\right)\right) \le \epsilon.$$

Now assume that we have

$$d_{TV}\left(\overline{g_{\sigma,q}} \circ \overline{h}^R\left(\overline{U}, t-1\right), \overline{g_{\sigma,q}} \circ \overline{\hat{h}_i}^R\left(\overline{U}, t-1\right)\right) \le t\epsilon. \tag{5}$$

We want to bound the total variation distance between $\overline{g_{\sigma,q}} \circ \overline{h}^R\left(\overline{U}, t\right)$ and $\overline{g_{\sigma,q}} \circ \overline{\hat{h}_i}^R\left(\overline{U}, t\right)$, which are defined as follows.

$$\overline{g_{\sigma,q}} \circ \overline{h}^R\left(\overline{U}, t\right) = \overline{g_{\sigma,q}} \circ f \circ \overline{g_{\sigma,s}}\left(\begin{bmatrix}\text{First}\left(\overline{h}^R\left(\overline{U}, t-1\right)\right) \\ \overline{u^{(t)}}\end{bmatrix}\right),$$

$$\overline{g_{\sigma,q}} \circ \overline{\hat{h}_i}^R\left(\overline{U}, t\right) = \overline{g_{\sigma,q}} \circ \hat{f}_i \circ \overline{g_{\sigma,s}}\left(\begin{bmatrix}\text{First}\left(\overline{\hat{h}_i}^R\left(\overline{U}, t-1\right)\right) \\ \overline{u^{(t)}}\end{bmatrix}\right).$$

From Lemma 29 we know that

$$d_{TV}\left(\text{First}\left(\overline{g_{\sigma,q}}\left(\overline{h}^R\left(\overline{U}, t-1\right)\right)\right), \text{First}\left(\overline{g_{\sigma,q}}\left(\overline{\hat{h}_i}^R\left(\overline{U}, t-1\right)\right)\right)\right)$$

$$\leq d_{TV}\left(\overline{g_{\sigma,q}}\left(\overline{h}^R\left(\overline{U}, t-1\right)\right), \overline{g_{\sigma,q}}\left(\overline{\hat{h}_i}^R\left(\overline{U}, t-1\right)\right)\right) \leq t\epsilon$$

It is easy to verify that $\text{First}\left(\overline{g_{\sigma,q}}\left(\overline{h}^R\left(\overline{U}, t-1\right)\right)\right) = \overline{g_{\sigma,q-1}}\left(\text{First}\left(\overline{h}^R\left(\overline{U}, t-1\right)\right)\right)$ because $\overline{g_{\sigma,q}}$ is a gaussian noise with covariance matrix equal to $\sigma^2 I_q$, where $I_q \in \mathbb{R}^{q \times q}$ is the identity matrix. Considering this fact and Lemma 30 we can write that

$$d_{TV}\left(\begin{bmatrix}\overline{g_{\sigma,q-1}}\left(\text{First}\left(\overline{h}^R\left(\overline{U}, t-1\right)\right)\right) \\ \overline{u^{(t)}}\end{bmatrix}, \begin{bmatrix}\overline{g_{\sigma,q-1}}\left(\text{First}\left(\overline{\hat{h}_i}^R\left(\overline{U}, t-1\right)\right)\right) \\ \overline{u^{(t)}}\end{bmatrix}\right)$$

$$\leq d_{TV}\left(\text{First}\left(\overline{g_{\sigma,q}}\left(\overline{h}^R\left(\overline{U}, t-1\right)\right)\right), \text{First}\left(\overline{g_{\sigma,q}}\left(\overline{\hat{h}_i}^R\left(\overline{U}, t-1\right)\right)\right)\right) \leq t\epsilon.$$

Applying data processing inequality for TV distance (i.e., Lemma 27) we can write that

$$d_{TV}\left(\begin{bmatrix}\overline{g_{\sigma,q-1}}\left(\text{First}\left(\overline{h}^R\left(\overline{U}, t-1\right)\right)\right) \\ \overline{g_{\sigma,p}}\left(\overline{u^{(t)}}\right)\end{bmatrix}, \begin{bmatrix}\overline{g_{\sigma,q-1}}\left(\text{First}\left(\overline{\hat{h}_i}^R\left(\overline{U}, t-1\right)\right)\right) \\ \overline{g_{\sigma,p}}\left(\overline{u^{(t)}}\right)\end{bmatrix}\right)$$

$$= d_{TV}\left(\overline{g_{\sigma,s}}\left(\begin{bmatrix}\text{First}\left(\overline{h}^R\left(\overline{U}, t-1\right)\right) \\ \overline{u^{(t)}}\end{bmatrix}\right), \overline{g_{\sigma,s}}\left(\begin{bmatrix}\text{First}\left(\overline{\hat{h}_i}^R\left(\overline{U}, t-1\right)\right) \\ \overline{u^{(t)}}\end{bmatrix}\right)\right) \leq t\epsilon. \tag{6}$$

Notice that $\begin{bmatrix}\text{First}\left(\overline{h}^R\left(\overline{U}, t-1\right)\right) & \overline{u^{(t)}}\end{bmatrix}^\top$ is in $\overline{\mathcal{X}_s}$. Since we know that $\overline{g_{\sigma,q}} \circ f \circ \overline{g_{\sigma,s}}$ and $\overline{g_{\sigma,q}} \circ \hat{f}_i \circ \overline{g_{\sigma,s}}$ are globally $\epsilon$-close on $\overline{\mathcal{X}_s}$, we can write that

$$d_{TV}\left(\overline{g_{\sigma,q}} \circ f \circ \overline{g_{\sigma,s}}\left(\begin{bmatrix}\text{First}\left(\overline{h}^R\left(\overline{U}, t-1\right)\right) \\ \overline{u^{(t)}}\end{bmatrix}\right), \overline{g_{\sigma,q}} \circ \hat{f}_i \circ \overline{g_{\sigma,s}}\left(\begin{bmatrix}\text{First}\left(\overline{h}^R\left(\overline{U}, t-1\right)\right) \\ \overline{u^{(t)}}\end{bmatrix}\right)\right) \leq \epsilon. \tag{7}$$

Moreover, from data processing inequality (i.e., Lemma 27) and Equation 6 we can conclude that

$$d_{TV}\left(\overline{g_{\sigma,q}} \circ \hat{f}_i \circ \overline{g_{\sigma,s}}\left(\begin{bmatrix}\text{First}\left(\overline{h}^R\left(\overline{U}, t-1\right)\right) \\ \overline{u^{(t)}}\end{bmatrix}\right), \overline{g_{\sigma,q}} \circ \hat{f}_i \circ \overline{g_{\sigma,s}}\left(\begin{bmatrix}\text{First}\left(\overline{\hat{h}_i}^R\left(\overline{U}, t-1\right)\right) \\ \overline{u^{(t)}}\end{bmatrix}\right)\right) \leq t\epsilon. \tag{8}$$

Finally, we can combine Equations 7 and 8 together with the triangle inequality for total variation distance to conclude that

$$d_{TV}\left(\overline{g_{\sigma,q}} \circ f \circ \overline{g_{\sigma,s}}\left(\begin{bmatrix}\text{First}\left(\overline{h}^R\left(\overline{U}, t-1\right)\right) \\ \overline{u^{(t)}}\end{bmatrix}\right), \overline{g_{\sigma,q}} \circ \hat{f}_i \circ \overline{g_{\sigma,s}}\left(\begin{bmatrix}\text{First}\left(\overline{\hat{h}_i}^R\left(\overline{U}, t-1\right)\right) \\ \overline{u^{(t)}}\end{bmatrix}\right)\right)$$

$$= d_{TV}\left(\overline{g_{\sigma,q}} \circ \overline{h}^R\left(\overline{U}, t\right), \overline{g_{\sigma,q}} \circ \overline{\hat{h}_i}^R\left(\overline{U}, t\right)\right) \leq (t+1)\epsilon.$$

718 So far, we have proved that for any input matrix $\overline{U} \in \overline{\Delta_{p \times T}}$ we have

$$d_{TV}\left(\overline{g_{\sigma,q}} \circ \overline{h}^R\left(\overline{U}, T-1\right), \overline{g_{\sigma,q}} \circ \widehat{\overline{h}}_i^R\left(\overline{U}, T-1\right)\right) \leq T\epsilon$$

719 By another application of Lemma 29 we can conclude that

$$d_{TV}\left(\text{Last}\left(\overline{g_{\sigma,q}} \circ \overline{h}^R\left(\overline{U}, T-1\right)\right), \text{Last}\left(\overline{g_{\sigma,q}} \circ \widehat{\overline{h}}_i^R\left(\overline{U}, T-1\right)\right)\right) \leq T\epsilon$$

720 We can have a similar argument to the first function and write the above equation as

$$d_{TV}\left(\overline{g_{\sigma,1}} \circ \text{Last}\left(h^R\left(\overline{U}, T-1\right)\right), \overline{g_{\sigma,1}} \circ \text{Last}\left(\hat{h}_i^R\left(\overline{U}, T-1\right)\right)\right) \leq T\epsilon.$$

721 This means that for every function $\overline{g_{\sigma,1}} \circ \text{Last}\left(\overline{h}^R\left(\overline{U}, T-1\right)\right)$ in $\overline{\mathcal{G}_{\sigma,1}} \circ \text{REC}[\overline{\mathcal{F}_\sigma}, T]$ there exists

722 a function $\hat{f}_i$ in $\mathcal{F}$ such that $\overline{g_{\sigma,1}} \circ \text{Last}\left(\overline{h}^R\left(\overline{U}, T-1\right)\right)$ and $\overline{g_{\sigma,1}} \circ \text{Last}\left(\widehat{\overline{h}}_i^R\left(\overline{U}, T-1\right)\right)$ are

723 globally $T\epsilon$-cover close to each other with respect to $\overline{\Delta_{p \times T}}$. Setting $\epsilon' = \epsilon/T$ we can conclude that

$$N_U\left(\epsilon, \overline{\mathcal{G}_\sigma} \circ \text{REC}[\overline{\mathcal{F}_\sigma}, T], \infty, d_{TV}^\infty, \overline{\Delta_{p \times T}}\right) \leq N_U\left(\frac{\epsilon}{T}, \overline{\mathcal{G}_{\sigma,q}} \circ \overline{\mathcal{F}_\sigma}, \infty, d_{TV}^\infty, \overline{\mathcal{X}_s}\right).$$

724 The proof of the bounded domains essentially follows the same steps as above but for inputs that are
725 bounded, i.e., inputs in $\overline{\Delta_{B,p \times T}}$ and $\overline{\mathcal{X}_{B,s}}$. $\qquad\square$

## D.2 A bound on the TV covering number of multi-layer noisy networks

727 From Theorem 25 and Theorem 23 we can get the following bound on the total variation covering
728 number of noisy multi-layer networks.

729 **Theorem 38** (TV Cover for Multi-Layer Noisy Neural Networks). *For every $\epsilon, \sigma \in (0,1)$ and every*
730 *well-defined class $\overline{MNET_\sigma}[p_0, p_k, w]$, we have*

$$\log N_U(\epsilon, \overline{\mathcal{G}_{\sigma,p_k}} \circ \overline{MNET_\sigma}[p_0, p_k, w], \infty, d_{TV}^\infty, \overline{\mathcal{X}_{0.5,p_0}})$$
$$= O\left(w \log\left(\frac{w}{\epsilon\sigma} \log\left(\frac{w}{\epsilon\sigma}\right)\right)\right) = \widetilde{O}\left(w \log\left(\frac{1}{\epsilon\sigma}\right)\right),$$

731 *where $\widetilde{O}$ hides logarithmic factors.*

732 *Proof.* Fix a choice of $p_1, \ldots, p_{k-1} \in \mathbb{N}$ and let $\overline{\mathcal{F}} = \underline{\text{NET}[p_{k-1}, p_k]} \circ \ldots \circ \overline{\mathcal{G}_\sigma} \circ \text{NET}[p_0, p_1] \circ \overline{\mathcal{G}_\sigma}$
733 be a class of multi-layer sigmoid neural networks in $\overline{MNET_\sigma}[p_0, p_k, w]$. Notice that

$$\overline{\mathcal{G}_\sigma} \circ \overline{\mathcal{F}} = \overline{\mathcal{G}_\sigma} \circ \text{NET}[p_{k-1}, p_k] \circ \ldots \circ \overline{\mathcal{G}_\sigma} \circ \text{NET}[p_0, p_1] \circ \overline{\mathcal{G}_\sigma}$$

734 and that the covering number of $\overline{\mathcal{G}_\sigma} \circ \overline{\mathcal{F}}$ with respect to $\overline{\mathcal{X}_{1,p_0}}$ is the same as the covering number of
735 $\overline{\mathcal{G}_\sigma} \circ \text{NET}[p_{k-1}, p_k] \circ \ldots \circ \overline{\mathcal{G}_\sigma} \circ \text{NET}[p_0, p_1]$ with respect to $\overline{\mathcal{G}_{\sigma,p_0}} \circ \overline{\mathcal{X}_{1,p_0}}$. From Theorem 25 we
736 know that for any $0 \leq i \leq k-1$ we can bound the covering number of $\overline{\mathcal{G}_\sigma} \circ \text{NET}[p_i, p_{i+1}]$ as

$$\log N_U\left(\epsilon, \overline{\mathcal{G}_\sigma} \circ \text{NET}[p_i, p_{i+1}], \infty, d_{TV}^\infty, \overline{\mathcal{G}_\sigma} \circ \overline{\mathcal{X}_{0.5,p_i}}\right)$$
$$\leq p_i(p_{i+1}+1) \log\left(30\frac{p_i^{5/2}\sqrt{\ln\left((5p_i - \epsilon\sigma)/(\epsilon\sigma)\right)}}{\epsilon^{3/2}\sigma^2} \ln\left(\frac{5p_i}{\epsilon\sigma}\right)\right).$$

737 Note that in Fathollah Pour and Ashtiani (2022) the above bound was originally stated as a bound
738 on the covering number of $\overline{\mathcal{G}_\sigma} \circ \text{NET}[p_i, p_{i+1}]$ with respect $\overline{\mathcal{G}_\sigma} \circ \overline{\mathcal{X}_{1,p_i}}$. However, we know that the
739 bound with respect to $\overline{\mathcal{G}_\sigma} \circ \overline{\mathcal{X}_{1,p_i}}$ is always an upper bound for the covering number with respect to
740 $\overline{\mathcal{G}_\sigma} \circ \overline{\mathcal{X}_{0.5,p_i}}$. If, instead of setting $B = 1$, we wanted to consider $B = 0.5$ as a bound on the domain,
741 the covering number bound would become only tighter in terms of constant factors. Considering the

 above facts and applying Theorem 23 recursively, we can write that

$$\log N_U\left(k\epsilon, \overline{\mathcal{G}_\sigma} \circ \overline{\mathcal{F}}, \infty, d_{TV}^\infty, \overline{\mathcal{X}_{0.5,p_0}}\right)$$

$$\leq \sum_{i=0}^{k-1} \log N_U\left(\epsilon, \overline{\mathcal{G}_\sigma} \circ \text{NET}[p_i, p_{i+1}], \infty, d_{TV}^\infty, \overline{\mathcal{G}_\sigma} \circ \overline{\mathcal{X}_{0.5,p_i}}\right)$$

$$\leq \sum_{i=0}^{k-1} p_i(p_{i+1}+1)\log\left(30\frac{p_i^{5/2}\sqrt{\ln\left((5p_i-\epsilon\sigma)/(\epsilon\sigma)\right)}}{\epsilon^{3/2}\sigma^2}\ln\left(\frac{5p_i}{\epsilon\sigma}\right)\right).$$

743 We can now set $\epsilon' = \epsilon/k$ and rewrite the above equation as

$$\log N_U\left(\epsilon, \overline{\mathcal{G}_\sigma} \circ \overline{\mathcal{F}}, \infty, d_{TV}^\infty, \overline{\mathcal{X}_{0.5,p_0}}\right)$$

$$\leq \sum_{i=0}^{k-1} p_i(p_{i+1}+1)\max_i\left\{\log\left(30\frac{p_i^{5/2}\sqrt{\ln\left((5p_i-\epsilon'\sigma)/(\epsilon'\sigma)\right)}}{\epsilon'^{3/2}\sigma^2}\ln\left(\frac{5p_i}{\epsilon'\sigma}\right)\right)\right\}$$

$$\leq w\max_i\left\{\log\left(30\frac{p_i^{5/2}\sqrt{\ln\left((5p_i-\epsilon'\sigma)/(\epsilon'\sigma)\right)}}{\epsilon'^{3/2}\sigma^2}\ln\left(\frac{5p_i}{\epsilon'\sigma}\right)\right)\right\}$$

$$\leq w\max_i\left\{\log\left(30\frac{p_i^{5/2}\sqrt{\ln\left(5p_i/(\epsilon'\sigma)\right)}}{\epsilon'^{3/2}\sigma^2}\ln\left(\frac{5p_i}{\epsilon'\sigma}\right)\right)\right\}$$

$$\leq w\max_i\left\{\log\left(30\frac{p_i^{5/2}\sqrt{5p_i/(\epsilon'\sigma)}}{\epsilon'^{3/2}\sigma^2}\ln\left(\frac{5p_i}{\epsilon'\sigma}\right)\right)\right\}$$

$$\leq w\max_i\left\{\log\left(30\sqrt{5}\frac{p_i^3}{\epsilon'^2\sigma^{3/2}}\ln\left(\frac{5p_i}{\epsilon'\sigma}\right)\right)\right\}.$$

744 Using the fact that $\epsilon, \sigma < 1$, we can simplify the above equation and write that

$$\log N_U\left(\epsilon, \overline{\mathcal{G}_\sigma} \circ \overline{\mathcal{F}}, \infty, d_{TV}^\infty, \overline{\mathcal{X}_{0.5,p_0}}\right)$$

$$\leq w\max_i\left\{\log\left((30\sqrt{5})^3\frac{p_i^3}{\epsilon'^3\sigma^3}\left(\ln\left(\frac{5p_i}{\epsilon'\sigma}\right)\right)^3\right)\right\}$$

$$\leq w\max_i\left\{3\log\left(30\sqrt{5}\frac{p_i}{\epsilon'\sigma}\ln\left(\frac{5p_i}{\epsilon'\sigma}\right)\right)\right\}$$

$$\leq w\max_i\left\{3\log\left(30\sqrt{5}\frac{kp_i}{\epsilon\sigma}\ln\left(\frac{5kp_i}{\epsilon\sigma}\right)\right)\right\}$$

$$\leq w\left(3\log\left(30\sqrt{5}\frac{w^2}{\epsilon\sigma}\ln\left(\frac{5w^2}{\epsilon\sigma}\right)\right)\right)$$

$$= O\left(w\log\left(\frac{w}{\epsilon\sigma}\ln\left(\frac{w}{\epsilon\sigma}\right)\right)\right) = \tilde{O}\left(w\log\left(\frac{1}{\epsilon\sigma}\right)\right),$$

745 where we used the fact that $k \leq w$ and $p_i \leq w$ for every $0 \leq i \leq k$. Now that we found an
746 upper bound on the covering number of $\overline{\mathcal{G}_\sigma} \circ \overline{\mathcal{F}}$ for a choice of $p_1, \ldots, p_{k-1}$, we can bound the
747 covering number of $\overline{\mathcal{G}_\sigma} \circ \overline{\text{MNET}_\sigma}[p_0, p_k, w]$. The number of different choices that we can have
748 for $p_1, \ldots, p_{k-1}$ is at most $w^{k-1}$ since we know that $\sum_{i=1}^k p_ip_{i-1} = w$ and therefore $p_i < w$ for
749 every $0 \leq i \leq k$. Therefore, we can simply take a union of the covering sets for each choice of
750 $p_0, \ldots, p_{k-1}$ as a covering set for $\overline{\mathcal{G}_\sigma} \circ \overline{\text{MNET}_\sigma}[p_0, p_k, w]$, which yields to the following covering
751 number bound.

$$\log N_U\left(\epsilon, \overline{\mathcal{G}_\sigma} \circ \overline{\text{MNET}_\sigma}[p_0, p_k, w], \infty, d_{TV}^\infty, \overline{\mathcal{X}_{0.5,p_0}}\right)$$

$$\leq \log w^k. N_U\left(\epsilon, \overline{\mathcal{G}_\sigma} \circ \overline{\mathcal{F}}, \infty, d_{TV}^\infty, \overline{\mathcal{G}_{\sigma,p_0}} \circ \overline{\mathcal{X}_{0.5,p_0}}\right)$$

$$\leq w\log w + \log N_U\left(\epsilon, \overline{\mathcal{G}_\sigma} \circ \overline{\mathcal{F}}, \infty, d_{TV}^\infty, \overline{\mathcal{G}_{\sigma,p_0}} \circ \overline{\mathcal{X}_{0.5,p_0}}\right) \qquad (k \leq w)$$

$$= O\left(w\log w + w\log\left(\frac{w}{\epsilon\sigma}\ln\left(\frac{w}{\epsilon\sigma}\right)\right)\right) = O\left(w\log\left(\frac{w}{\epsilon\sigma}\ln\left(\frac{w}{\epsilon\sigma}\right)\right)\right) = \tilde{O}\left(w\log\left(\frac{1}{\epsilon\sigma}\right)\right),$$

752 $\qquad\qquad\qquad\qquad\qquad\qquad\qquad\qquad\qquad\qquad\qquad\qquad\qquad\qquad\qquad\qquad\qquad\qquad\qquad\qquad\qquad\square$

### D.3 Proof of Theorem 26

*Proof.* We know that

$$\overline{\text{MNET}_\sigma}[p_0, p_k, w] = \bigcup \text{NET}[p_{k-1}, p_k] \circ \ldots \circ \overline{\mathcal{G}_\sigma} \circ \text{NET}[p_1, p_2] \circ \overline{\mathcal{G}_\sigma} \circ \text{NET}[p_0, p_1] \circ \overline{\mathcal{G}_\sigma}.$$

Define $\overline{\mathcal{F}} = \bigcup \text{NET}[p_{k-1}, p_k] \circ \ldots \circ \overline{\mathcal{G}_\sigma} \circ \text{NET}[p_1, p_2] \circ \overline{\mathcal{G}_\sigma} \circ \text{NET}[p_0, p_1]$ and note that $\overline{\mathcal{F}} \circ \overline{\mathcal{G}_\sigma} = \overline{\mathcal{F}_\sigma} = \overline{\text{MNET}_\sigma}[p_0, p_k, w]$. Therefore, we can use Theorem 24 to write that

$$N_U\left(\epsilon, \overline{\mathcal{G}_\sigma} \circ \text{REC}[\overline{\text{MNET}_\sigma}[p_0, p_k, w], T], \infty, d_{TV}^\infty, \overline{\Delta_{0.5, p \times T}}\right)$$
$$= N_U\left(\epsilon, \overline{\mathcal{G}_\sigma} \circ \text{REC}[\overline{\mathcal{F}_\sigma}, T], \infty, d_{TV}^\infty, \overline{\Delta_{0.5, p \times T}}\right)$$
$$\leq N_U\left(\frac{\epsilon}{T}, \overline{\mathcal{G}_\sigma} \circ \overline{\mathcal{F}_\sigma}, \infty, d_{TV}^\infty, \overline{\mathcal{X}_{0.5, s}}\right)$$
$$= N_U\left(\frac{\epsilon}{T}, \overline{\mathcal{G}} \circ \overline{\text{MNET}_\sigma}[p_0, p_k, w], \infty, d_{TV}^\infty, \overline{\mathcal{X}_{0.5, s}}\right).$$

We know of a bound on the covering number of $\overline{\mathcal{G}_\sigma} \circ \overline{\text{MNET}_\sigma}[p_0, p_k, w]$ from Theorem 38. Using this bound we can rewrite the above equation as

$$N_U\left(\epsilon, \overline{\mathcal{G}_\sigma} \circ \text{REC}[\overline{\text{MNET}_\sigma}[p_0, p_k, w], T], \infty, d_{TV}^\infty, \overline{\Delta_{0.5, p \times T}}\right)$$
$$\leq N_U\left(\frac{\epsilon}{T}, \overline{\mathcal{G}} \circ \overline{\text{MNET}_\sigma}[p_0, p_k, w], \infty, d_{TV}^\infty, \overline{\mathcal{X}_{0.5, s}}\right)$$
$$= O\left(w \log\left(\frac{wT}{\epsilon\sigma} \ln\left(\frac{wT}{\epsilon\sigma}\right)\right)\right) = \widetilde{O}\left(w \log\left(\frac{T}{\epsilon\sigma}\right)\right).$$

$\square$

### D.4 Proof of Theorem 19

*Proof.* From Theorem 43 we can write that

$$\mathbb{E}_{(x,y) \sim \mathcal{D}}\left[l_\gamma(\hat{f}, x, y)\right]$$

$$\leq \inf_{f \in \mathcal{F}} \mathbb{E}_{(x,y) \sim \mathcal{D}}\left[l_\gamma(f, x, y)\right] + 2 \inf_{\epsilon \in [0, 1/2]} \left\{2\left[4\epsilon + \frac{12}{\sqrt{m}} \int_\epsilon^{1/2} \sqrt{\ln N_U(\gamma\nu, \mathcal{F}, m, \|.\|_2^{\ell_2})} \, d\nu\right]\right\} + 6\sqrt{\frac{\ln(2/\delta)}{2m}}$$

$$\leq \inf_{f \in \mathcal{F}} \mathbb{E}_{(x,y) \sim \mathcal{D}}\left[l_\gamma(f, x, y)\right] + 2\left[8\epsilon + \frac{24}{\sqrt{m}} \int_\epsilon^{1/2} \sqrt{\ln N_U(\gamma\nu, \mathcal{F}, m, \|.\|_2^{\ell_2})} \, d\nu\right] + 6\sqrt{\frac{\ln(2/\delta)}{2m}} \qquad (\forall \epsilon \in [0, 1/2])$$

$$\leq \inf_{f \in \mathcal{F}} \mathbb{E}_{(x,y) \sim \mathcal{D}}\left[l_\gamma(f, x, y)\right] + 16\epsilon + \frac{24}{\sqrt{m}} \sqrt{\ln N_U(\gamma\epsilon, \mathcal{F}, m, \|.\|_2^{\ell_2})} + 6\sqrt{\frac{\ln(2/\delta)}{2m}},$$

where we have used the fact that the integral is over $[0, 1/2]$ and the covering number decreases monotonically with $\epsilon$. $\square$

### D.5 Proof of Theorem 15

We are now ready to state the proof of the upper bound on the sample complexity of PAC learning noisy recurrent neural networks with respect to the ramp loss.

*Proof.* From Theorem 19 we know that if we choose algorithm $\mathcal{A}$ such that for every distribution over $[-1/2, 1/2]^{p \times T} \times \{-1, 1\}$ and any input $S$ of $m$ i.i.d. samples from $\mathcal{D}$ it outputs $\mathcal{A}(S) = \hat{h} = \arg\min_{h \in \mathcal{H}_w} \frac{1}{|S|} \sum_{(x,y) \in S} l_\gamma(h, x, y)$, then with probability at least $1 - \delta$ we have

$$\mathbb{E}_{(U,y) \sim \mathcal{D}}\left[l_\gamma(\hat{h}, U, y)\right]$$

$$\leq \inf_{h \in \mathcal{H}_w} \mathbb{E}_{(U,y) \sim \mathcal{D}}\left[l_\gamma(h, U, y)\right] + 16\epsilon + \frac{24}{\sqrt{m}} \sqrt{\log N_U(\gamma\epsilon, \mathcal{H}_w, m, \|.\|_2^{\ell_2})} + 6\sqrt{\frac{\log(2/\delta)}{2m}}. \qquad (9)$$

770 We know that $\overline{\mathcal{Q}_w}$ is a class of functions from $[-1/2, 1/2]^{p \times T}$ to $[-1/2, 1/2]$. We also know that
771 $\|x\|_2^{\ell_2} \leq \|x\|_2^{\infty}$ for every $x$. We can now use Theorem 22 to turn the bound on the covering
772 number of $\overline{\mathcal{Q}_w}$ to a bound on the covering number of $\mathcal{E}(\overline{\mathcal{Q}_w})$. Note that Theorem 22 is stated
773 for functions with outputs in $[-B, B]$ and $\overline{\mathcal{Q}_w} = \overline{\mathcal{G}_\sigma} \circ \text{REC}[\overline{\text{MNET}_\sigma}[p_0, p_k, w], T]$ outputs values
774 in $\overline{\mathcal{G}_{\sigma, p_k}} \circ \overline{\mathcal{X}_{0.5, p_k}}$. However, $\overline{\mathcal{G}_{\sigma, p_k}}$ is a class of zero mean Gaussian random variables that are
775 independent of the output of $\text{REC}[\overline{\text{MNET}_\sigma}[p_0, p_k, w], T]$ and, therefore, they do not change the
776 expectation and the covering number bound for $\mathcal{E}(\overline{\mathcal{Q}_w})$ would be the same as the covering number
777 bound for $\mathcal{E}\left(\text{REC}[\overline{\text{MNET}_\sigma}[p_0, p_k, w], T]\right)$. Thus we know that

$$N_U(\gamma\epsilon, \mathcal{H}_w, m, \|.\|_2^{\ell_2}) \leq N_U(\gamma\epsilon, \mathcal{H}_w, m, \|.\|_2^{\infty}) \leq N_U(\gamma\epsilon, \overline{\mathcal{Q}_w}, \infty, d_{TV}^{\infty}, \overline{\Delta_{0.5, p \times T}}).$$

778 We can, therefore, rewrite Equation 9 as follows.

$$\mathbb{E}_{(U,y) \sim \mathcal{D}}\left[l_\gamma(\hat{h}, U, y)\right]$$

$$\leq \inf_{h \in \mathcal{H}_w} \mathbb{E}_{(U,y) \sim \mathcal{D}}\left[l_\gamma(h, U, y)\right] + 16\epsilon + \frac{24}{\sqrt{m}}\sqrt{\log N_U(\gamma\epsilon, \overline{\mathcal{Q}_2}, \infty, d_{TV}^{\infty}, \overline{\Delta_{0.5, p \times T}})} + 6\sqrt{\frac{\log(2/\delta)}{2m}}.$$
(10)

779 Therefore, if we find $m$ such that $\frac{1}{\sqrt{m}}\sqrt{\log N_U(\gamma\epsilon, \overline{\mathcal{Q}_w}, \infty, d_{TV}^{\infty}, \overline{\Delta_{p \times T}})} = O(\epsilon)$ and $\sqrt{\frac{\log(1/\delta)}{m}} =$
780 $O(\epsilon)$ then we can guarantee $\mathbb{E}_{(U,y) \sim \mathcal{D}}\left[l_\gamma(\hat{h}, U, y)\right] \leq \inf_{h \in \mathcal{H}_w} \mathbb{E}_{(U,y) \sim \mathcal{D}}\left[l_\gamma(h, U, y)\right] + O(\epsilon)$

781 We know of a covering number bound for $\overline{\mathcal{Q}_w}$ from Theorem 26 which is as follows.

$$\log N_U\left(\epsilon, \overline{\mathcal{Q}_w}, \infty, d_{TV}^{\infty}, \overline{\Delta_{0.5, p \times T}}\right) = O\left(w \log\left(\frac{wT}{\epsilon\sigma} \ln\left(\frac{wT}{\epsilon\sigma}\right)\right)\right).$$

782 We can thus write that

$$\sqrt{\frac{\log N_U(\gamma\epsilon, \overline{\mathcal{Q}_w}, \infty, d_{TV}^{\infty}, \overline{\Delta_{0.5, p \times T}})}{m}} = O(\epsilon) \Leftrightarrow m = O\left(\frac{1}{\epsilon^2} w \log\left(\frac{wT}{\epsilon\sigma} \ln\left(\frac{wT}{\epsilon\sigma}\right)\right)\right)$$

783 Moreover, if we want $\sqrt{\frac{\log(1/\delta)}{m}} = O(\epsilon)$ then we should have $m = O\left(\frac{\log(1/\delta)}{\epsilon^2}\right)$. Combining the
784 above results, we can conclude that

$$m_{\mathcal{H}_w}(\epsilon, \delta) = O\left(\frac{w \log\left(\frac{wT}{\epsilon\sigma} \ln\left(\frac{wT}{\epsilon\sigma}\right)\right) + \log(1/\delta)}{\epsilon^2}\right) = \widetilde{O}\left(\frac{w \log\left(\frac{T}{\sigma}\right) + \log(1/\delta)}{\epsilon^2}\right).$$

785 samples is sufficient to conclude that with probability at least $1 - \delta$ we have $\mathbb{E}_{(U,y) \sim \mathcal{D}}\left[l_\gamma(\hat{h}, U, y)\right] \leq$
786 $\inf_{h \in \mathcal{H}_w} \mathbb{E}_{(U,y) \sim \mathcal{D}}\left[l_\gamma(h, U, y)\right] + O(\epsilon)$, which implies PAC learning $\mathcal{H}_w$ with respect to ramp loss
787 with a sample complexity of $m_{\mathcal{H}_w}(\epsilon, \delta)$. □

## E PAC learning and covering number bounds

789 In this section, we discuss how we can find a bound on the sample complexity of PAC learning a class
790 of functions with respect to ramp loss from a bound on its covering number. Particularly, we show
791 how to use a bound on covering number to find the number of samples required to ensure uniform
792 convergence with respect to ramp loss. We then connect the uniform convergence results to PAC
793 learning and find the minimum number of samples required to guarantee PAC learning with respect
794 to ramp loss.

795 We start by defining uniform convergence (with respect to ramp loss).

796 **Definition 39** (Uniform Convergence with Respect to Ramp Loss). *Let $\mathcal{F}$ be a class of functions*
797 *from $\mathcal{X}$ to $\mathbb{R}$. We say that $\mathcal{F}$ has uniform convergence property with respect to ramp loss with margin*
798 *parameter $\gamma > 0$ if there exists some function $m : (0, 1)^2 \to \mathbb{N}$ such that for every distribution*
799 *$\mathcal{D}$ over $\mathcal{X} \times \{-1, 1\}$ and every $\epsilon, \delta \in (0, 1)$, if $S$ is a set of $m(\epsilon, \delta)$ i.i.d. samples from $\mathcal{D}$, then*
800 *with probability at least $1 - \delta$ (over the randomness of $S$) for every function $f \in \mathcal{F}$ we have*
801 *$\left|\mathbb{E}_{(x,y) \sim \mathcal{D}}[l_\gamma(f, x, y)] - \frac{1}{|S|}\sum_{(x,y) \in S} l_\gamma(f, x, y)\right| \leq \epsilon$.*

The *sample complexity* of uniform convergence for class $\mathcal{F}$ is denoted by $m_{\mathcal{F}}^{\text{UC}}(\epsilon, \delta)$, which is the minimum number of samples required to guarantee uniform convergence for $\mathcal{F}$. We now show that uniform convergence implies PAC learning (with respect to ramp loss).

**Lemma 40.** *Let $\mathcal{F}$ be a class of functions from $\mathcal{X}$ to $\mathbb{R}$ that satisfies uniform convergence property with respect to ramp loss. Then for any $(\epsilon, \delta) \in (0, 1)$, we have $m_{\mathcal{F}}(\epsilon, \delta) \leq m_{\mathcal{F}}^{UC}(\epsilon/2, \delta)$, i.e., there exists an algorithm $\mathcal{A}$ such that for any distribution $\mathcal{D}$ over $\mathcal{X} \times \{-1, 1\}$ and any $(\epsilon, \delta) \in (0, 1)$, if $S$ is a set of $m \geq m_{\mathcal{F}}^{UC}(\epsilon/2, \delta)$ i.i.d. samples from $\mathcal{D}$, then with probability at least $1 - \delta$, we have that $\mathbb{E}[(x, y) \sim \mathcal{D}] \, l_\gamma(\mathcal{A}(S), x, y) \leq \inf_{f \in \mathcal{F}} \mathbb{E}_{(x,y) \sim \mathcal{D}}[l_\gamma(f, x, y)] + \epsilon$.*

*Proof.* Let $\mathcal{A}$ be an algorithm that outputs the function in $\mathcal{F}$ that has the minimum empirical loss, i.e., $\mathcal{A}(S) = \arg\min_{f \in \mathcal{F}} \frac{1}{|S|} \sum_{(x,y) \in S} l_\gamma(f, x, y)$. Since $S$ is a set of $m \geq m_{\mathcal{F}}^{\text{UC}}(\epsilon/2, \delta)$ samples, we know that with probability at least $1 - \delta$ we have $\left| \mathbb{E}_{(x,y) \sim \mathcal{D}}[l_\gamma(f, x, y)] - \frac{1}{|S|} \sum_{(x,y) \in S} l_\gamma(f, x, y) \right| \leq \epsilon/2$ for every $f \in \mathcal{F}$. Let $\hat{f} = \mathcal{A}(S)$. Then for every $f \in \mathcal{F}$ we can write that

$$\mathbb{E}_{(x,y) \sim \mathcal{D}}\left[ l_\gamma(\hat{f}, x, y) \right] \leq \frac{1}{|S|} \sum_{(x,y) \in S} l_\gamma(\hat{f}, x, y) + \frac{\epsilon}{2} \leq \frac{1}{|S|} \sum_{(x,y) \in S} l_\gamma(f, x, y) + \frac{\epsilon}{2}$$

$$\leq \mathbb{E}_{(x,y) \in \mathcal{D}}\left[ l_\gamma(f, x, y) \right] + \frac{\epsilon}{2} + \frac{\epsilon}{2} = \mathbb{E}_{(x,y) \in \mathcal{D}}\left[ l_\gamma(f, x, y) \right] + \epsilon.$$

This implies that with $m \geq m_{\mathcal{F}}^{\text{UC}}(\epsilon/2, \delta)$ i.i.d. samples we can guarantee PAC learning with respect to ramp loss with parameters $\epsilon$ and $\delta$. In other words, we have $m_{\mathcal{F}}(\epsilon, \delta) \leq m_{\mathcal{F}}^{\text{UC}}(\epsilon/2, \delta)$. $\qquad\square$

The following theorem tells us that we can relate the bound on the covering number of a class of functions to the uniform convergence property for that class. The proof relies on bounding the Rademacher complexity of the class by a bound on its covering number (Dudley, 2010) and then relating the bound on the Rademacher complexity to uniform convergence property. See Shalev-Shwartz and Ben-David (2014) and Mohri et al. (2018) for a more detailed discussion and proof.

**Theorem 41.** *Let $\mathcal{F}$ be a class of functions from $\mathcal{X}$ to $\mathbb{R}$ and $\mathcal{F}_\gamma = \{f_\gamma : \mathcal{X} \times \{-1, 1\} \to [0, 1] \mid f_\gamma(x, y) = r_\gamma(-f(x).y), f \in \mathcal{F}\}$ be the class of its composition with ramp loss. Let $\mathcal{D}$ be a distribution over $\mathcal{X} \times \{-1, 1\}$ and $S \sim \mathcal{D}^m$ be an i.i.d. sample of size $m$. Then, for any $\delta \in (0, 1)$ with probability at least $1 - \delta$ (over the randomness of $S$) for every $f \in \mathcal{F}$ we have*

$$\mathbb{E}_{(x,y) \sim \mathcal{D}}[l_\gamma(f, x, y)]$$
$$\leq \frac{1}{|S|} \sum_{(x,y) \in S} l_\gamma(f, x, y) + \inf_{\epsilon \in [0, 1/2]} \left\{ 2 \left[ 4\epsilon + \frac{12}{\sqrt{m}} \int_\epsilon^{1/2} \sqrt{\log N_U(\nu, \mathcal{F}_\gamma, m, \|.\|_2^{\ell_2})} \, d\nu \right] \right\} + 3 \sqrt{\frac{\log(2/\delta)}{2m}}.$$

It is only left to find a bound on the covering number of $\mathcal{F}_\gamma$ from a bound on the covering number of $\mathcal{F}$. The following lemma helps us finding this bound.

**Lemma 42** (From Covering Number of $\mathcal{F}$ to Covering Number of $\mathcal{F}_\gamma$). *Let $\mathcal{F}$ be a class of functions from $\mathcal{X}$ to $\mathbb{R}$ and $\mathcal{F}_\gamma = \{f_\gamma : \mathcal{X} \times \{-1, 1\} \to [0, 1] \mid f_\gamma(x, y) = r_\gamma(-f(x).y), f \in \mathcal{F}\}$ be the class of its composition with ramp loss. Then we have*

$$N_U(\epsilon, \mathcal{F}_\gamma, m, \|.\|_2^{\ell_2}) \leq N_U(\gamma\epsilon, \mathcal{F}, m, \|.\|_2^{\ell_2}).$$

*Proof.* First, it is easy to verify that $r_\gamma$ (with respect to the first input) is a Lipschitz continuous function with respect to $\|.\|_2$ with Lipschitz factors of $1/\gamma$; see e.g., section A.2 in Bartlett et al. (2017).

Fix an input set $S = \{(x_1, y_1), \ldots, (x_m, y_m)\} \subset \mathcal{X} \times \mathcal{Y}$ and let $C = \{\hat{f}_{i|S} \mid \hat{f}_i \in \mathcal{F}, i \in [r]\}$ be an $(\gamma\epsilon)$-cover for $\mathcal{F}_{|S}$. For the simplicity of notation, we denote the composition of $\hat{f}_i$ with ramp loss by $\hat{f}_{\gamma,i}$. Now, we prove that $C_\gamma = \{\hat{f}_{\gamma,i_{|S}} \mid \hat{f}_{\gamma,i} \in \mathcal{F}_\gamma, i \in [r]\}$ is also an $\epsilon$-cover for $\mathcal{F}_{\gamma|S}$.

Given any $f \in \mathcal{F}$, there exists $\hat{f}_{i|S} \in C$ such that

$$\left\| (\hat{f}_i(x_1), \ldots, \hat{f}_i(x_m)) - (f(x_1), \ldots, f(x_m)) \right\|_2^{\ell_2} \leq \gamma\epsilon.$$

We can then write that

$$\left\| (\hat{f}_{\gamma,i}(x_1), \ldots, \hat{f}_{\gamma,i}(x_m)) - (f_\gamma(x_1), \ldots, f_\gamma(x_m)) \right\|_2^{\ell_2}$$

$$= \sqrt{\frac{1}{m} \sum_{k=1}^m \left( \hat{f}_{\gamma,i}(x_k) - f_\gamma(x_k) \right)^2} \tag{11}$$

$$= \sqrt{\frac{1}{m} \sum_{k=1}^m \left( r_\gamma \left( -\hat{f}_i(x_k).y_k \right) - r_\gamma(-f(x_k).y_k) \right)^2}.$$

From the Lipschitz continuity of $r_\gamma(x)$ we can conclude that for any $(x,y) \in \mathcal{X} \times \mathcal{Y}$,

$$\left| r_\gamma(-f(x).y) - r_\gamma(-\hat{f}_i(x).y) \right| \leq \frac{1}{\gamma} \left| \hat{f}_i(x) - f(x) \right|.$$

Taking the above equation into account, we can rewrite Equation 11 as

$$\left\| (\hat{f}_{\gamma,i}(x_1), \ldots, \hat{f}_{\gamma,i}(x_m)) - (f_\gamma(x_1), \ldots, f_\gamma(x_m)) \right\|_2^{\ell_2}$$

$$\leq \frac{1}{\gamma} \sqrt{\frac{1}{m} \sum_{k=1}^m \left( (\hat{f}_i(x_k) - f(x_k)) \right)^2}$$

$$\leq \frac{1}{\gamma} \left\| (\hat{f}_i(x_1), \ldots, \hat{f}_i(x_m)) - (f(x_1), \ldots, f(x_m)) \right\|_2^{\ell_2}$$

$$\leq \frac{1}{\gamma} \gamma \epsilon$$

$$\leq \epsilon.$$

In other words, for any $f_{\gamma|S} \in \mathcal{F}_{\gamma|S}$ there exists $\hat{f}_{\gamma,i_{|S}} \in S$ such that $\left\| \hat{f}_{\gamma,i_{|S}} - f_{\gamma|S} \right\|_2^{\ell_2} \leq \epsilon$ and, therefore, $C_\gamma$ is an $\epsilon$-cover for $\mathcal{F}_{\gamma|S}$ and the result follows. $\qquad\square$

We can now combine Theorem 41, Lemma 40, and Lemma 42 to state the following theorem, which implies that we can relate a bound on the covering number of a class $\mathcal{F}$ to PAC learning $\mathcal{F}$ with respect to ramp loss.

**Theorem 43.** *Let $\mathcal{F}$ be a class of functions from $\mathcal{X}$ to $\mathbb{R}$. There exists an algorithm $\mathcal{A}$ with the following property: For every distribution $\mathcal{D}$ over $\mathcal{X} \times \{-1, 1\}$ and every $\delta \in (0,1)$, if $S$ is a set of $m$ i.i.d. samples from $\mathcal{D}$, the algorithm outputs a hypothesis $f = \mathcal{A}(S)$ such that with probability at least $1 - \delta$ (over the randomness of $S$ and $\mathcal{A}$) we have*

$$\mathbb{E}_{(x,y)\sim\mathcal{D}} [l_\gamma(f,x,y)]$$

$$\leq \inf_{f\in\mathcal{F}} \mathbb{E}_{(x,y)\sim\mathcal{D}} [l_\gamma(f,x,y)] + 2 \inf_{\epsilon\in[0,1/2]} \left\{ 2 \left[ 4\epsilon + \frac{12}{\sqrt{m}} \int_\epsilon^{1/2} \sqrt{\log N_U(\nu, \mathcal{F}_\gamma, m, \|.\|_2^{\ell_2})} \, d\nu \right] \right\} + 6\sqrt{\frac{\log(2/\delta)}{2m}}.$$

*Proof.* From Theorem 41 we know that for every $f \in \mathcal{F}$ with probability at least $1 - \delta$ we have

$$\mathbb{E}_{(x,y)\sim\mathcal{D}} [l_\gamma(f,x,y)]$$

$$\leq \frac{1}{|S|} \sum_{(x,y)\in S} l_\gamma(f,x,y) + \inf_{\epsilon\in[0,1/2]} \left\{ 2 \left[ 4\epsilon + \frac{12}{\sqrt{m}} \int_\epsilon^{1/2} \sqrt{\log N_U(\nu, \mathcal{F}_\gamma, m, \|.\|_2^{\ell_2})} \, d\nu \right] \right\} + 3\sqrt{\frac{\log(2/\delta)}{2m}}.$$

850 Lemma 40 suggests that if we choose algorithm $\mathcal{A}$ such that $\mathcal{A}(S) = \hat{f} = $
851 $\arg\min_{f \in \mathcal{F}} \frac{1}{|S|} \sum_{(x,y) \in S} l_\gamma(f, x, y)$ then for any $\hat{f} \in \mathcal{F}$ with probability at least $1 - \delta$ we have

$$\mathbb{E}_{(x,y) \sim \mathcal{D}} \left[ l_\gamma(\hat{f}, x, y) \right]$$

$$\leq \frac{1}{|S|} \sum_{(x,y) \in S} l_\gamma(\hat{f}, x, y) + \inf_{\epsilon \in [0,1/2]} \left\{ 2 \left[ 4\epsilon + \frac{12}{\sqrt{m}} \int_\epsilon^{1/2} \sqrt{\log N_U(\nu, \mathcal{F}_\gamma, m, \|.\|_2^{\ell_2})} \, d\nu \right] \right\} + 3 \sqrt{\frac{\log(2/\delta)}{2m}}$$

$$\leq \frac{1}{|S|} \sum_{(x,y) \in S} l_\gamma(f, x, y) + \inf_{\epsilon \in [0,1/2]} \left\{ 2 \left[ 4\epsilon + \frac{12}{\sqrt{m}} \int_\epsilon^{1/2} \sqrt{\log N_U(\gamma\nu, \mathcal{F}, m, \|.\|_2^{\ell_2})} \, d\nu \right] \right\} + 3 \sqrt{\frac{\log(2/\delta)}{2m}}$$

$$\leq \mathbb{E}_{(x,y) \sim \mathcal{D}} \left[ l_\gamma(f, x, y) \right] + 2 \inf_{\epsilon \in [0,1/2]} \left\{ 2 \left[ 4\epsilon + \frac{12}{\sqrt{m}} \int_\epsilon^{1/2} \sqrt{\log N_U(\gamma\nu, \mathcal{F}, m, \|.\|_2^{\ell_2})} \, d\nu \right] \right\} + 6 \sqrt{\frac{\log(2/\delta)}{2m}}$$

$$\leq \inf_{f \in \mathcal{F}} \mathbb{E}_{(x,y) \sim \mathcal{D}} \left[ l_\gamma(f, x, y) \right] + 2 \inf_{\epsilon \in [0,1/2]} \left\{ 2 \left[ 4\epsilon + \frac{12}{\sqrt{m}} \int_\epsilon^{1/2} \sqrt{\log N_U(\gamma\nu, \mathcal{F}, m, \|.\|_2^{\ell_2})} \, d\nu \right] \right\} + 6 \sqrt{\frac{\log(2/\delta)}{2m}}.$$

852 $\qquad\qquad\qquad\qquad\qquad\qquad\qquad\qquad\qquad\qquad\qquad\qquad\qquad\qquad\qquad\qquad\qquad\qquad\qquad\qquad\qquad$ $\square$

853 In Appendix D we use the above theorem together with an approximation of the right hand side of the
854 above inequality to find an upper bound on the sample complexity of PAC learning noisy recurrent
855 neural networks with respect to ramp loss.

# F   Missing proof from Section 7

## F.1   Proof of Theorem 22

858 *Proof.* Let $S = \{U_1, \ldots, U_m\} \subset \mathbb{R}^{p \times T}$ be an input set and define $\overline{S} = \{\overline{U_1}, \ldots, \overline{U_m}\} \subset \overline{\Delta_{p \times T}}$.
859 Let $C = \{\overline{\hat{f}_{1|\overline{S}}}, \ldots, \overline{\hat{f}_{r|\overline{S}}} \mid \overline{\hat{f}_r} \in \overline{\mathcal{F}}, i \in [r]\}$ be an $\epsilon$-cover for $\overline{\mathcal{F}}_{|\overline{S}}$ with respect to $d_{TV}^\infty$. Denote
860 $\mathcal{H} = \mathcal{E}(\overline{\mathcal{F}})$ and let $\hat{\mathcal{H}} = \left\{ \hat{h}_i(x) = \mathbb{E}_{\overline{\hat{f}_i}} \left[ \overline{\hat{f}_i}(x) \right] \mid i \in [r] \right\} \subset \mathcal{E}(\overline{\mathcal{F}})$ be a new set of non-random
861 function.
862 Given any random function $\overline{f} \in \overline{\mathcal{F}}$ and considering the fact that $C$ is an $\epsilon$-cover for $\overline{\mathcal{F}}_{|\overline{S}}$ we know
863 there exists $\overline{\hat{f}_i}$, $i \in [r]$ such that

$$d_{TV}^\infty \left( \overline{\hat{f}_{i|\overline{S}}}, \overline{f}_{|\overline{S}} \right) = d_{TV}^\infty \left( (\overline{\hat{f}_i}(\overline{U_1}), \ldots, \overline{\hat{f}_i}(\overline{U_m})), (\overline{f}(\overline{U_1}), \ldots, \overline{f}(\overline{U_m})) \right) \leq \epsilon.$$

864 From the above equation we can conclude that for any $k \in [m]$ we have $d_{TV} \left( \overline{\hat{f}_i}(\overline{U_k}), \overline{f}(\overline{U_k}) \right) \leq \epsilon$.
865 Further, for the corresponding $h, \hat{h}_i \in \mathcal{E}(\overline{\mathcal{F}})$, we know that

$$\hat{h}_i(U_k) = \mathbb{E}_{\overline{\hat{f}_i}} \left[ \overline{\hat{f}_i}(\overline{U_k}) \right] = \int_{\mathbb{R}^d} x \mathscr{D}(\overline{\hat{f}_i}(\overline{U_k}))(x) dx,$$

$$h(U_k) = \mathbb{E}_{\overline{f}} \left[ \overline{f}(\overline{U_k}) \right] = \int_{\mathbb{R}^d} x \mathscr{D}(\overline{f}(\overline{U_k}))(x) dx.$$

866 Denote $I = \mathscr{D}(\overline{f}(\overline{U_k}))$ and $\hat{I} = \mathscr{D}(\overline{\hat{f}_i}(\overline{U_k}))$. Define two new density functions $I_{diff}$ and $\hat{I}_{diff}$ as

$$I_{diff}(x) = \begin{cases} \dfrac{I(x) - \hat{I}(x)}{d_{TV}(I, \hat{I})} & I(x) \geq \hat{I}(x) \\[4mm] 0 & \text{otherwise,} \end{cases}$$

$$\hat{I}_{diff}(x) = \begin{cases} \dfrac{\hat{I}(x) - I(x)}{d_{TV}(I, \hat{I})} & \hat{I}(x) \geq I(x) \\[4mm] 0 & \text{otherwise.} \end{cases}$$

867    Also, we define $I_{min}$ as

$$I_{min}(x) = \frac{\min\{I(x), \hat{I}(x)\}}{\int \min\{I(x), \hat{I}(x)\}dx} = \frac{\min\{I(x), \hat{I}(x)\}}{1 - d_{TV}(I, \hat{I})}.$$

868    We can verify that

$$I(x) = \left(1 - d_{TV}(I, \hat{I})\right) I_{min}(x) + d_{TV}(I, \hat{I}).I_{diff}(x)$$

$$\hat{I}(x) = \left(1 - d_{TV}(I, \hat{I})\right) I_{min}(x) + d_{TV}(I, \hat{I}).\hat{I}_{diff}(x).$$

869    We then find the $\ell_2$ distance between $\hat{h}_i(U_k)$ and $h(U_k)$ by

$$\left\|\hat{h}_i(U_k) - h(U_k)\right\|_2$$

$$= \left\|\int_{\mathbb{R}^d} x\hat{I}(x)dx - \int_{\mathbb{R}^d} xI(x)dx\right\|_2$$

$$= \left\|\int_{\mathbb{R}^d} x\left[\left(1 - d_{TV}(I, \hat{I})\right) I_{min}(x) + d_{TV}(I, \hat{I}).\hat{I}_{diff}(x)\right]\right.$$

$$\left. -x\left[\left(1 - d_{TV}(I, \hat{I})\right) I_{min}(x) + d_{TV}(I, \hat{I}).I_{diff}(x)\right] dx\right\|_2$$

$$= \left\|\int_{\mathbb{R}^d} xd_{TV}(I, \hat{I})\left[\hat{I}_{diff}(x) - I_{diff}(x)\right] dx\right\|_2$$

$$= d_{TV}(I, \hat{I})\left\|\int_{\mathbb{R}^d} x\left[\hat{I}_{diff}(x) - I_{diff}(x)\right] dx\right\|_2$$

$$\leq 2B\sqrt{q}\, d_{TV}\left(\overline{f}(\overline{U_k}), \overline{\hat{f}_i}(\overline{U_k})\right) \qquad\qquad \text{(Bounded domain } [-B, B]^q \text{ and triangle inequality)}$$

$$\leq 2B\epsilon\sqrt{q}.$$

870    Since this result holds for any $k \in [m]$, we have

$$\|\hat{h}_{i|S} - h_{|S}\|_2^{\ell_2} = \sqrt{\frac{1}{m}\sum_{k=1}^{m}\left\|\hat{h}_i(U_k) - h(U_k)\right\|_2^2}$$

$$\leq \sqrt{\frac{1}{m}\sum_{k=1}^{m}(2B\sqrt{q})^2\left(d_{TV}\left(\overline{f}(\overline{U_k}), \overline{\hat{f}_i}(\overline{U_k})\right)\right)^2} \leq 2B\sqrt{q}\sqrt{\frac{1}{m}\sum_{k=1}^{m}\epsilon^2} \leq 2B\epsilon\sqrt{q}.$$

871    Therefore, $\hat{\mathcal{H}}_{|S}$ is a $2B\epsilon\sqrt{q}$ cover for $\mathcal{H}_{|S}$ with respect to $\|.\|_2^{\ell_2}$ and $|\hat{\mathcal{H}}_{|S}| = r$. This holds for any
872    subset $S$ of $\mathbb{R}^{p \times T}$ with $|S| = m$. Therefore,

$$N_U(2B\epsilon\sqrt{q}, \mathcal{E}(\overline{\mathcal{F}}), m, \|.\|_2^{\ell_2}) \leq N_U(\epsilon, \overline{\mathcal{F}}, m, d_{TV}^\infty, \overline{\Delta_{p \times T}}) \leq N_U(\epsilon, \overline{\mathcal{F}}, \infty, d_{TV}^\infty, \overline{\Delta_{p \times T}}).$$

873    $\qquad\qquad\qquad\qquad\qquad\qquad\qquad\qquad\qquad\qquad\qquad\qquad\qquad\qquad\qquad\qquad\qquad\qquad\qquad\qquad\qquad$ $\square$