# OpenReview forum: "On the Role of Noise in the Sample Complexity of Learning Recurrent Neural Networks: Exponential Gaps for Long Sequences"
_NeurIPS.cc/2023/Conference — NeurIPS 2023 poster_

### Official Review · Reviewer_9rJ1 · 2023-07-06

**Soundness:** 4 excellent
**Presentation:** 4 excellent
**Contribution:** 4 excellent
**Rating:** 7
**Confidence:** 4

**Summary:**

The authors consider the class of noisy multi-layered sigmoid recurrent neural networks, noisy meaning that noise is added to the output of each neuron in the network. They prove that the sample complexity of noisy is significantly better than non-noisy one with one clean upper bound and one clean lower bound.

**Strengths:**

1. Clear writing
2. Well organised presentation
3. Reasonable assumptions on an important model
4. Studied multilayer neural networks, making a complete story.
5. Interesting technique TV Cover, handy for composition classes

**Weaknesses:**

Here are some minor weaknesses.

1. The authors could have discussed more about the practical implications. It occurs to me that adding noise might be one of the keys to next generation AI now that deep learning is currently struggling with sampling complexity.
2. Would be better if there could be discussion about expressive power.
3. sigmoid not RELU.
4. lack of experimental results

**Questions:**

How much does noise affect expressive power?

**Limitations:**

The authors adequately addressed most of the limitations.

---

> ### Author Rebuttal · Authors · 2023-08-09
>
> We thank the reviewer for their suggestions and comments. We discuss the questions and concerns mentioned by the reviewer in the following.
>
> **Response to weaknesses.** We agree with the reviewer that practical result can back up the results of the paper and we hope to have this as a future work. Also, extending the results to ReLU networks rather than the sigmoid function is a future direction for this work that can increase its generality.
>
> **Response to Question.** This is a very good question. We want to mention that as discussed in the paper our results also work in the regime where we have negligble amounts of noise and the noise is only required to enable the analysis. So if we are considering the current computers with finite precision, the noisy and non-noisy networks perform almost the same when implemented on them and we can get exponential improvements in the sample complexity without losing any expressive power. A related discussion is also included in Lines 61-66 and 210-213. We think analyzing the expressive power for the regime where the noise is not negligible and therefore these two classes may have different performances is an interesting future direction.

---

> > ### Comment · Reviewer_9rJ1 · 2023-08-10
> >
> > Very good, makes sense.

---

### Official Review · Reviewer_KR7t · 2023-07-14

**Soundness:** 3 good
**Presentation:** 3 good
**Contribution:** 3 good
**Rating:** 6
**Confidence:** 3

**Summary:**

This work studies the sample complexity of PAC learning noisy recurrent neural networks with respect to the ramp loss. Noisy recurrent networks are defined as multi-layered feed forward networks with sigmoid activation where independent mean-zero Gaussian noise is added to the activation.

The main result is that one can learn these noisy networks with sample complexity $\tilde{O}(\frac{w\log(T/\sigma) + \log(1/\delta)}{\epsilon^2})$ (Theorem 15). Though different in terms of loss function and noise assumption with prior work, the bound notably avoids any dependence on the norms of the weights, which was common in upper bounds in prior work. These results are much more intriguing once contrasted to its non-noisy counterpart, i.e., the fact that non-noisy PAC learning has a lower bound of $\Omega(\frac{wT+\log(1/\delta)}{\epsilon^2})$ (Theorem 10), which exhibits linear dependence on $T$ as opposed to $\log(T)$ as in the noisy case.

The techniques used to prove the upper bound may be of independent interest to the PAC learning community. The authors use techniques derived from Fathollah Pour and Ashtiani (2022) to study covering numbers of random hypotheses and extend these tools so that their analysis is based on uniform covering number with respect to total variation distance.

**Strengths:**

- The paper shows an interesting contrast between learning noisy vs. clean recurrent neural networks. There is surprisingly an exponential gap in sample complexity w.r.t. $T$ and that noisy networks are easier to learn. This gives way to the suggestion that learning clean networks can "bypass" the existing hardness result by assuming there is an infinitesimally small noise $\sigma$ close to 0 for the clean network, allowing the actual sample complexity upper bound be smaller in case of finite precision machines.
- The proof techniques are interesting as the paper works with random hypotheses and establishing sample complexity via covering numbers under such randomness.

**Weaknesses:**

- The motivation to study noisy recurrent neural networks, as it is defined in this paper, is unclear to me. The formulation is similar to a noisy dynamical system but it is unclear why such noise assumption would be natural here for neural networks.

**Questions:**

- On one hand, learning noisy networks being easier than learning clean networks is surprising since it's noisier. Yet, it may be the case that, by adding noise (smaller signal-to-noise ratio), it becomes easier to learn as more hypotheses may seem to plausible (as if blurring an image). Is this why the noise is helping the learner?
- What is the main reason for studying the ramp loss?

**Limitations:**

The paper is clear with limitations and comparison to prior/future work.

---

> ### Author Rebuttal · Authors · 2023-08-09
>
> We thank the reviewer for their assessment and feedback. We address the concerns and questions raised by the reviewer in the following.
>
> **Response to Weaknesses.** Using noise in training neural networks is a natural heuristic to avoid overfitting and has been used in various scenarios (e.g., in denoising autoencoders, drop-out noise, noisy RNNs, etc.). For RNNs, the issue of overfitting is more serious especially for learning from long sequences. Our study can quantify the effect of noise for generalization in RNNs.
>
> But perhaps more importantly, we use noisy RNNs as a "proof technique". As discussed in Lines 61-66 and 210-213, in practice we can set the amount of noise so negligible that when implemented on a device with finite precision the network performs almost similar to the natural non-noisy version, without sacrificing the performance. In other words, our analysis shows that there is a stark difference between finite precision machines and true real-valued machines when it comes to training RNNs.
>
> On a more technical level, adding noise to the network makes it possible to use the TV covering number analysis which comes with a useful composition theorem. It makes it possible to use new techniques and exploit the fact that the same function in being applied recursively and get a sample complexity bound that is exponentially smaller than the non-noisy version.
>
> **Response to Question 1.** Although it may be intuitive that noise makes the class less complex, one of our contributions is actually proving a bound on the complexity of noisy recurrent models by using new techniques and analysis for random functions. It is worth mentioning that we are analyzing the sample complexity of PAC learning this class; we want to find the smallest number of samples required to make sure that, for any data generating distribution, the error of the trained/returned network $\hat{f}\in\overline{\mathcal{F}}$ is comparable with the smallest expected error that is achievable by a noisy network from the same class $\overline{\mathcal{F}}$, i.e., $\arg\min_{f\in\overline{\mathcal{F}}}\mathbb{E}{l^{\gamma}(f,x,y)}$. It may seem intuitive that by adding noise to a single classifier $f$ we can hope to reduce its overfitting and bound the difference between its empirical and expected error. However, it is not obvious and immediate that the expected error of returned classifier $\hat{f}$ is close to the minimum possible expected error over the class of noisy networks, i.e., whether $\mathbb{E}{l^{\gamma}(\hat{f},x,y)} \leq \min_{f\in\overline{\mathcal{F}}}\mathbb{E}{l^{\gamma}(f,x,y)} + \epsilon$ with high probability, and if we can guarantee PAC learning with smaller sample complexity. Our main contribution is a novel technique for studying the complexity of noisy RNNs and proving that the sample complexity of PAC learning them is logarithmic in $T$.
>
> **Response to Question 2.** As mentioned in Line 139, the main features of the ramp loss that we use are the Lipschitzness and boundedness, which are common in the literature of neural networks for analyzing the sample complexity. Otherwise, if the loss function is unbounded, it would be impossible to guarantee generalization for networks with unbounded weights. Our sample complexity analysis for noisy networks is more general and works for any loss function that has the above properties. We chose to use ramp loss because we wanted to have concrete theorem statements and to be able to contrast the upper bound of noisy network with the lower bound of the non-noisy networks.

---

> > ### Comment · Reviewer_KR7t · 2023-08-14
> > **Reply to Authors**
> >
> > I have read the rebuttal. It would certainly be interesting to see if the upper bound of the noisy network can be generalized to deal with any Lipschitz bounded function w.r.t. related parameters like Lipschitzness.
> > Thank you for the response.

---

> > > ### Author Response · Authors · 2023-08-14
> > > **Reply to Reviewer KR7t**
> > >
> > > Thanks for reading our rebuttal and the response. We would like to elaborate more on why our upper bound holds for any bounded Lipschitz loss function. Lemma 42 in Appendix E is the main result for translating the covering number of the network to that of its composition with the loss. The only property of ramp loss that is used in this proof is its Lipschitzness, i.e., Line 838. Replacing ramp loss with any other loss with Lipschitz continuity of $L$ will lead to a similar result, i.e., if we let $\mathcal{F}\_{L}$ to be the composition of $\mathcal{F}$ with the Lipschitz loss, then we have $N_U(\epsilon, \mathcal{F}_{L}, m,||.||_2^{\ell_2}) \leq N_U(\epsilon/L, \mathcal{F}, m,||.||_2^{\ell_2})$. Consequently, in the proof of Theorem 15 we would only need to use a covering number with accuracy of $\epsilon/L$ instead of $\epsilon\gamma$ between Lines 782 and 783. This results in a logarithmic dependence on $L$ in the final sample complexity, i.e., Theorem 15 for a loss with Lipschitz continuous factor of $L$ would state a sample complexity of $\tilde{O}\left(\frac{w\log (\frac{TL}{\sigma})+\log(1/\delta)}{\epsilon^2} \right)$.

---

### Official Review · Reviewer_hjDk · 2023-07-15

**Soundness:** 4 excellent
**Presentation:** 4 excellent
**Contribution:** 3 good
**Rating:** 6
**Confidence:** 3

**Summary:**

This paper studies learning recurrent neural networks with sigmoid activations, where a small amount of noise of magnitude sigma is added to each layer. This paper shows that there is a log(T/sigma) scaling of the sample complexity with the number of recurrent compositions.

 This is surprising because, without the noise, there is a linear in T scaling of the sample complexity. Furthermore, the quantity of noise can be very small, since the log(1/sigma) dependence is mild.

**Strengths:**

The paper is notationally heavy, but is clearly written and was therefore easy to follow. The comparison with past works was good, and I felt that the main result was surprising.

**Weaknesses:**


From a technical point of view:
* The proof of Theorem 10 (the sample complexity lower bound for learning with the Ramp Loss) is effectively the same as Koiran and Sontag (1998), who proved it for the 0-1 loss. The main observation is that neural networks with sigmoid activations can be forced to output approximately 0-1 valued functions by rescaling the final layer weights. And therefore the ramp loss is approximately equal to the 0-1 loss.

* Most of the steps in the proof of Theorem 26 draw heavily on the work of "Benefits of Additive Noise in Composing Classes with Bounded Capacity" by Fathollah Pour and Ashtiani, 2022. The main technical novelty seems to be in Theorem 24, which gives an inequality for the TV covering number under composition. Did I understand correctly that this is the main new element?

**Questions:**

* Is the noise $\mathcal{G}_{\sigma}$ in Definition 13 reused, or fresh noise on each layer?


* There seem to be some minor technical issues with the proof of Theorem 10 as stated. In the proof of Lemma 11 (restated as Lemma 32), the authors on line 604 define $z$. The authors write the definition of $z$ as
$$z = \min_b \arg\max_{0 < x < 1/2} P[|Last(b^R(U,T-1))| \geq x] \geq 1-\eta.$$
This is nonstandard notation and it is unclear how to parse it. The authors state that "intuitively, z is the largest possible value such that "
$$P[-z < Last(b^R(U,T-1)) < z] < \eta$$ Considering this and the remainder of the proof, I am assuming that the authors meant
$$z = \max \\{x \in (0,1/2) :  P[|Last(b^R(U,T-1))| \geq x] \geq 1-\eta \mbox{ for all } b\\}.$$
However, $z$ may not be well-defined because the maximum might not exist. Consider the network $b$ that sends any input to $Last(b^R(U,T-1)) = 0$ almost surely. Then no $x \in (0,1/2)$ is valid for the above problem -- we would have to take $x = 0$, which would lead to an issue with the proof since then $z = 0$.

* How does the construction for Theorem 10 break if sigma=1/T^C magnitude noise is added for some large constant C? Your Theorem 26 would predict T*polylog(T) sample complexity, ao the construction in Theorem 10 should break, but I don't see exactly how. Could you please provide some intuition?

Typos:
* Appendix C, Proof of Theorem 10, have [l]^{\gamma} and [l]^{0-1}, when it should be [l^{gamma}] and [l^{0-1}]
* Statement of Theorem 23, display math between lines 311 and 312. Why is there ". N_1" in the expression? Is it supposed to be times N_1?

**Limitations:**

Yes

---

> ### Author Rebuttal · Authors · 2023-08-09
>
> We would like to thank the reviewer for their careful assessment of the paper and we appreciate their helpful comments. The questions and concerns mentioned by the reviewer are discussed in the following.
>
> **Response to Weaknesses.** Yes, the key technical contribution is to prove a general result on the covering number of recurrent models based on the recurring class, and a sample complexity upper bound that is logarithmic in $T$ (hence, the surprising exponential gap). On the other hand, most (if not all) of the techniques used in the literature (for non-noisy networks) are based on "unfolding the recurrence" and result in a linear dependence on $T$. In fact, as we showed in the lower bound, for non-noisy networks, getting a (super-)linear upper bound is inevitable.
>
> We would like to add that simply using the framework of Fathollah Pour and Ashtiani (2022) will again give a linear dependence on $T$. Therefore, we had to use a new proof technique to be able to exploit the fact that in RNNs, a fixed function is being reused recursively. With this and other nuances in the proof (e.g., handling the new data that is inputted to the network in every recurrence) we were able to prove the sub-linear (logarithmic upper) bound.
>
> **Response to Questions 1.** The noise is i.i.d. in and fresh sample is used in each layer.
>
> **Response to Questions 2.**  We thank the reviewer for bringing this to our attention! In fact, there is a mistake in the current proof but can be fixed easily. In short, instead of defining a general value of $c$, we can define a specific value $c_b$ for any given network $b$ to resolve the issue mentioned by the reviewer.
>
> To see this, we should have defined $z_b$ specifically for any network $b$ as $z_b = \sup \\{ 0\leq x<1/2:\mathbb{P}{\left|\text{Last}{b^R(U,T-1)}\right|  \geq x} \geq 1-\eta \\}$. For any $b$ and its corresponding $f \in \mathcal{F_w}$ if $z_b>0$ we can multiply the last row of the weight matrix of the last layer by $c_b=\phi^{-1}(\gamma)/\phi^{-1}(z_b)$. In case $z_b=0$, we do not need to change any weights and we let $c_b=1$. We can conclude for the same function $b$ and the corresponding $h_b = \text{Last}(b^R(U,T-1)) \in \mathcal{H_w}$ we have $\\mathbb{E}_\{(U,y)\\sim \\mathcal{D}}{l_\{\\gamma} (h\_b,U,y)} \\leq \\mathbb{E}_\{(U,y)\\sim \\mathcal{D}}{l^{0-1}(f,U,y)} + \\eta$ as desired because the ramp loss and the $0-1$ loss output the same value with probability more than $1-\eta$. Therefore, the result of Lemma 11 and Theorem 10 are correct and still hold. We make sure to modify the proof of Lemma 11 and make it exact in the next versions of this paper.
>
> **Response to Questions 3.** We hope we have understood the question correctly and if the reviewer is asking why if we set $\sigma$ to be inverse exponentially in $T$, then the upper bound in Theorem 15 becomes $T*polylog T$ but Theorem 10 suggests a lower bound of $\Omega(T)$. We want to emphasize that this is not a contradiction and the construction in Theorem 10 does not break. We would only have a loose upper bound that is not better than the lower bound in Theorem 10. The intuition behind having the loose upper bound when $\sigma\rightarrow 0$ is that we are analyzing the covering number with respect to TV distance which we show is sufficient for learning but it is not necessary in general. Intuitively, the noise is smoothing out the output and makes sure that with a small change in the weights of network the output distributions do not become very different, i.e., have small TV distance. However, when $\sigma\rightarrow 0$, the output distributions are almost close to the Dirac delta measures and the TV distance is almost equal to $1$, so we do not quite get the benefits that we wanted from adding noise. We showed that the sample complexity has mild logarithmic dependence on $1/\sigma$, so for moderately small values of $\sigma$ we would still get a tighter generalization bound for noisy networks. However, when $\sigma$ is set to be so small, one would lose the benefits of additive noise. In this regime, one can basically use the basic (super-)linear bounds for non-noisy RNNs.
>
> **Typos.** Yes, in the statement of Theorem 23 it is supposed to be times $N_1$. We thank the reviewer for noticing the typos and make sure that we will fix it in future versions of the paper.

---

> > ### Comment · Reviewer_hjDk · 2023-08-14
> > **Response**
> >
> > Thanks for answering my questions. I'm happy to keep my score.

---

### Official Review · Reviewer_Ei2R · 2023-07-16

**Soundness:** 3 good
**Presentation:** 3 good
**Contribution:** 3 good
**Rating:** 5
**Confidence:** 3

**Summary:**

In this paper, the authors consider a class of noisy recurrent neural networks under the ramp loss setting, and prove that the noisy class can be learned with $O(w \log (T/\sigma))$, where $w$ is the width, $T$ is the length of the sequence, and $\sigma$ is noise variance.

The derived results demonstrate the sample effiency when compared to the standard noiseless case.
The proof is based on the result of covering number of a class of random maps under the TV distance.

**Strengths:**

- the sample complexity $O(w \log (T/\sigma))$ is proved for such noise RNN under the ramp loss
- the covering number under the TV distance is given

**Weaknesses:**

- The motivation of using ramp loss is unclear to me and quite weak. For example, ramp loss is non-convex and not commonly used in practice. I understand that the ramp loss will be quite close to the 0-1 loss, and thus the derivation would be relatively easier, e.g., Lemma 11. Nevertheless, it decreases the technical difficulty as well as the motivation.

- The used techique is from (Fathollah Pour and Ashtiani 2022) on the covering number of a class of random maps under the TV distance. Their results have already applied to shallow and deeper NNs. The contribution appears not very significant under a somehow RNN with ramp loss. When checking the proof in a high level way, the key part, Lemma 11, aims to build the connection between the ramp loss and the 0-1 loss.

- According to the derived sample complexity, it shows that, under a larger noise level, this noisy class can be learned with fewer samples. It makes sense because noise (gradually become the main component) can be easier to be learned than the data. However, this result somethimes is not enough in my view. Because a larger noise injection would lead to a worse performance. Solely adding noise result in nothing. I think this requires more detailed discussion.

**Questions:**

See the above

---

> ### Author Rebuttal · Authors · 2023-08-09
>
> We would like to thank the reviewer for their comments and feedback. In the following, we address the concerns and questions mentioned in the review.
>
> **Response to Weaknesses 1.** As we mentioned in Line 139, the only features of the ramp loss that we use to derive the upper bound are its Lipschitzness and boundedness. Our results are more general and work for any other loss that has these properties. It is also common in the literature to assume these two assumptions for the loss function when the generalization of neural networks are analyzed theoretically. Otherwise, if the loss function is unbounded, it would be impossible to guarantee generalization without making additional assumptions on the norms of the weights of network.
>
> For the lower bound, we used the specific choice of ramp loss to be concrete in our theorem statements and to be able to contrast it with the upper bound. In fact, the constructions in Lemma 11 and, consequently, the lower bound in Theorem 10 only rely on the above two properties and an additional property that the loss function $l(z)$ converges to $0$ as $z$ goes to $\infty$ and converges to $1$ as $z$ goes to $-\infty$, which is again natural in the analysis of neural networks.
>
> **Response to Weaknesses 2.** We would like to emphasize that Theorem 15 (main upper bound result) is based on the result of Theorem 24 which is novel and not present in Fathollah Pour and Ashtiani (2022) (FA22). As we discussed in Lines 312-331, simply using the result of FA22 will give a loose (linear) upper bound in terms of the length of the sequence. Therefore, the bound would not be better than those of the existing literature that simply "unfold the recurrence" and analyze the recurrent network as a larger and more complex class. On the other hand, we prove that with a more careful analysis of the noisy function it is possible to take into account the fact that the same fixed function is being applied recursively. Using the new analysis that we offer for any recurrent model, we conclude tighter generalization bound for RNNs which is logarithmic in $T$. When contrasted with our (linear) lower bound, we get the exponential gap as a surprising result. There are other nuances in the proofs that are reflected in the (relatively long) supplementary material.
>
> **Response to Weaknesses 3.** Proving a sample complexity bound for noisy RNNs that is logarithmic in $T$ is one of the main contributions of our paper and it was not, to the best of our knowledge, present in the literature before. We agree that intuitively, it is true and not surprising that adding noise reduces the complexity of the class of networks and results in tighter generalization bounds. However, what is surprising is the fact that even a negligible amount of noise is enough to enable the noisy analysis. For example if the neural network is implemented on a device with finite precision, then one can set $\sigma=10^{-240}$ and get a generalization bound which is logarithmic in $T$, without sacrificing any accuracy. These discussion are also included in Lines 61-66 and 210-213.

---

> > ### Comment · Reviewer_Ei2R · 2023-08-11
> >
> > Thanks for the authors' response.
> >
> > For the ramp loss, the authors mentioned that, this paper only requires the classification loss to be Lipchitz continuous and bounded. Actually it excludes the commonly used cross-entropy loss, hinge loss.
> >
> > The developed technique is beneficial to the recurrent models, and I'm wondering that the conclusion (e.g., noise helps sample efficiency) can still hold for general architectures? e.g., MLP, CNN, Transformer.

---

> > > ### Author Response · Authors · 2023-08-15
> > > **Reply to Reviewer Ei2R**
> > >
> > > Thanks for your response. Regarding the unbounded loss functions such as cross-entropy loss or hinge loss, we would like to draw the reviewer's attention to an important distinction between two scenarios: (i) where these loss functions are used as "surrogate losses" but the goal is bounding the classification (0-1) loss or (ii) where the actual goal is having a small expected value for these unbounded losses.
> > >
> > > The first scenario is common in the literature, and in fact our generalization bound remains valid in this case. To see this, note that the ramp loss is upper bounded by these surrogate losses. Therefore, a small empirical value of the surrogate loss also implies a small empirical ramp loss. On the other hand, we guarantee that the expected and empirical values of the ramp loss are close to each other. Therefore, we can conclude that the expected error of ramp loss is smaller than or close to the empirical value of the surrogate losses. We also know that the 0-1 loss is always upper bounded by the ramp loss. Therefore, bounding the sample complexity of generalization for ramp loss implies that the expected value of classification loss is not larger than the empirical value of these surrogate losses.
> > >
> > > For the second scenario it is impossible to bound the gap between the expected and empirical values of unbounded losses without making extra assumptions about the data distribution or the network, e.g., bounding the weights. Intuitively, this is because even a single mistake on a low probable input point can have a detrimental effect on population loss.
> > >
> > > We focused on the ramp loss to be able to demonstrate the gap between the lower and the upper bounds. Otherwise, just like the first scenario, our upper bound can be used to bound the expected 0-1 error when the empirical value of the hinge loss or cross entropy loss is small.
> > >
> > > Regarding the second question, we want to emphasize that the main message of our paper is that noise can exponentially reduce the dependency of sample complexity on sequence length in RNNs (even for negligible amounts of noise), which is a surprising result and it is based on a novel technique for noisy recurrent models. If the question is about using MLP, CNN, etc. as the recurring class in the RNN, our result indeed is with respect to MLP (with no bound on the weights) as the recurring block. The extension to CNN, Transformer, etc., is an exiting future direction. If the question concerns other architectures in general, we think that since the noise is beneficial in MLPs and RNNs, it would be a good future direction to apply noisy analysis to other networks as well.

---

> > > > ### Comment · Reviewer_Ei2R · 2023-08-15
> > > >
> > > > thanks for the response.

---

### Decision · Program_Chairs · 2023-09-21

**Decision:**

Accept (poster)

**Comment:**

This paper studies the sample complexity of learning recurrent neural networks. The main claim is that when some noise is added to the output of each neuron, the sample complexity scales logarithmically in the sequence length T instead of linearly in the case of no noise. Although usually neural networks do not have any noise added, there has been interest in the setting with stochastic outputs for various reasons (e.g. related to generalization/pac-bayes theory). Overall the reviewers had a positive view of this contribution, and I recommend acceptance.